# Notch dimerization and gene dosage are important for normal heart development, intestinal stem cell maintenance, and splenic marginal zone B-cell homeostasis during mite infestation

Francis M. Kobia[1☯¤], Kristina Preusse[1☯], Quanhui Dai[1,2], Nicholas Weaver[3], Matthew R. Hass[1], Praneet Chaturvedi[1], Sarah J. Stein[4], Warren S. Pear[4], Zhenyu Yuan[5], Rhett A. Kovall[5], Yi Kuang[1], Natanel Eafergen[6], David Sprinzak[6], Brian Gebelein[1], Eric W. Brunskill[1], Raphael Kopan[1]*

1 Division of Developmental Biology, Department of Pediatrics, University of Cincinnati College of Medicine and Cincinnati Children's Hospital Medical Center, Cincinnati, Ohio, United States of America, 2 State Key Laboratory of Genetic Engineering, School of Life Sciences, Fudan University, Shanghai, China, 3 Immunology Graduate Program, University of Cincinnati College of Medicine, Cincinnati, Ohio, United States of America, 4 Department of Pathology and Laboratory Medicine, Abramson Family Cancer Research Institute, Perelman School of Medicine at the University of Pennsylvania, Philadelphia, Pennsylvania, United States of America, 5 Department of Molecular Genetics, Biochemistry and Microbiology, University of Cincinnati College of Medicine, Cincinnati, Ohio, United States of America, 6 School of Neurobiology, Biochemistry, and Biophysics, The George S. Wise Faculty of Life Sciences Tel Aviv University, Tel Aviv, Israel

☯ These authors contributed equally to this work.
¤ Current address: Mount Kenya University, Department of Research and Innovation, Thika, Kenya
* Raphael.Kopan@CCHMC.ORG

**Data Availability Statement:** All relevant data are in the paper and its Supplementary Data files.

## Abstract

Cooperative DNA binding is a key feature of transcriptional regulation. Here we examined the role of cooperativity in Notch signaling by CRISPR-mediated engineering of mice in which neither Notch1 nor Notch2 can homo- or heterodimerize, essential for cooperative binding to sequence-paired sites (SPS) located near many Notch-regulated genes. Although most known Notch-dependent phenotypes were unaffected in Notch1/2 dimer–deficient mice, a subset of tissues proved highly sensitive to loss of cooperativity. These phenotypes include heart development, compromised viability in combination with low gene dose, and the gut, developing ulcerative colitis in response to 1% dextran sulfate sodium (DSS). The most striking phenotypes—gender imbalance and splenic marginal zone B-cell lymphoma—emerged in combination with gene dose reduction or when challenged by chronic fur mite infestation. This study highlights the role of the environment in malignancy and colitis and is consistent with Notch-dependent anti-parasite immune responses being compromised in Notch dimer–deficient animals.

Sequencing data files were submitted to the Gene Expression Omnibus repository (GEO; https://www.ncbi.nlm.nih.gov/geo/; GSE149992).

**Funding:** Cell sorting was supported by NIH S10OD023410 and 01DK106225 to the CCHMC Research Flow Cytometry Core. BG and DS were supported at a grant from the National Science Foundation and the Binational Science Foundation (NSF/BSF -1715822). SJS was supported by NIH T32CA009140 and the American Cancer Society PF-15-065-01-TBG. WSP was supported by NIH R01 CA215518 to WSP. RK, FMK, KP, NW, and EB were supported by MIH R01 CA163353 to RK. RK and QD were supported by the William K. Schubert Endowed Chair at Cincinnati Children's Hospital Medical Center. QD receives a small stipend from the Chinese government. The funders had no role in study design, data collection and analysis, decision to publish, or preparation of the manuscript.

**Competing interests:** The authors have declared that no competing interests exist.

**Abbreviations:** ANK, Ankyrin repeat; ATAC-seq, Assay for Transposase Accessible Chromatin sequencing; ChIP, chromatin immunoprecipitation; CSL, CBF1/Suppressor of Hairless/LAG-1; D/AM, complementing halves of DAM; DAM, DNA adenine methylase; DHS, DNAse hypersensitive; DN, double negative (CD4$^-$ CD4$^-$) T cells; DP, double positive (CD4$^+$ CD8$^+$) T cells; DSS, dextran sulfate sodium; E, embryonic day; E(spl), enhancer of split; EMSA, electrophoretic mobility shift assay; F0, filial generation 0; FACS, fluorescence-activated cell sorting; FoB, follicular B-cell; FVB, Friend virus B–susceptible strain; H/E, hematoxylin/eosin; HDM, house dust mite; Hes, Hairy and Enhancer of Split; IgG, immunoglobulin G; IgM, immunoglobulin M; ISC, intestinal stem cell; Klf, Krüppel-like factor; LPS, lipopolysaccharide; Maml, Mastermind-like; MZB, marginal zone B-cell; MZP, MZB precursor; N1, Notch1; N1RA, Notch1 Arg$^{1974}$Ala; N2, Notch2; N2RA, Notch2 Arg$^{1934}$Ala; NICD, Notch intracellular domain; NRARP, Notch-regulated ankyrin repeat protein; NTC, Notch transcription complex; P, postnatal day; PEST, Pro-Glu-Ser-Thr rich domain; PP, post-permethrin; RBPj, recombinant binding protein for immunoglobulin Kappa j region; SMZL, splenic marginal zone B-cell lymphoma; SP, single positive (CD4$^-$ CD8$^+$ or CD4$^+$ CD$^-$) T cells; SPS, sequence-paired site; T2, type 2 cell; T-ALL, T cell acute lymphoblastic leukemia; Th2, T helper 2; TSS, transcription start site; VSD, ventricular septal defect; WT, wild-type.

## Introduction

The evolutionarily conserved Notch receptors and ligands influence metazoan development and adult tissue homeostasis by directly translating an intercellular interaction into intracellular transcriptional outputs that control cell fate, proliferation, differentiation, and apoptosis [1–3]. Mammals possess 4 Notch receptors (N1 to N4) and 5 Delta/Jagged ligands; all of which are Type I transmembrane proteins. The Notch pathway stands out relative to other signaling pathways in lacking signal amplification: Canonical Notch signaling is initiated when a ligand on one cell engages a Notch receptor on a neighboring cell. This interaction unfolds the receptor's juxtamembrane region enabling cleavage by the metalloprotease ADAM10. The truncated, cell-membrane-bound polypeptide is then cleaved by the γ-secretase complex freeing the Notch intracellular domain (NICD), which subsequently translocates into the nucleus [2, 4]. NICD associates with the DNA-binding protein CSL (CBF1/Suppressor of Hairless/LAG-1, also known as recombinant binding protein for immunoglobulin Kappa j region (RBPj) in vertebrates) and recruits the coactivator Mastermind-like (Maml), thereby assembling a Notch transcription complex (NTC) that activates Notch target gene expression [4–6].

The Notch pathway plays complex and context-dependent roles during development and adult tissue homeostasis. Perturbations in the Notch pathway are associated with developmental syndromes [7] and cancers [8, 9]. For example, *Notch1* (N1) promotes T-cell development [10–12], whereas *Notch2* (N2) is indispensable for marginal zone B-cell (MZB) development [13–20]. Accordingly, elevated Notch1 signaling is oncogenic in T cells driving acute lymphoblastic leukemia (T-ALL) [21, 22], whereas increased Notch2 signaling is associated with splenic MZB transformation [23–25]. Inversely, when Notch signals promote differentiation, the pathway can have a tumor suppressor function with diminished Notch signaling being associated with cancer [26, 27].

How can the Notch pathway control multiple, dissimilar outcomes? Because each Notch receptor is consumed as it generates a signal and cannot be reused, signal strength has emerged as a key in controlling outcome. Indeed, some Notch-dependent processes are exquisitely sensitive to dosage and manifest both haploinsufficient and triplomutant effects in tissues such as the fly wing [1, 28]. In mammalian tissues, signal strength, defined as the sum of NICD released from all ligand-bound Notch receptors on the cell surface, is a far more important determinant of Notch signaling outcomes than NICD composition (i.e., N1ICD versus N2ICD) [20, 29, 30]. Several mechanisms have been found to modulate signal strength, including receptor glycosylation [31, 32], force-generating ligand endocytosis [33, 34], the contact interface between cells [35], and the ratio of receptor/ligand constitution within the cell [36]. More recently, ligand-dependent signal dynamics were demonstrated to be another key determinant of signaling outcomes [37].

Interestingly, NICD can assemble into dimeric, cooperative NTCs [38] on sequence-paired sites (SPSs), first described in the regulatory regions of the *Drosophila* enhancer of split [E(spl)] locus [39]. SPSs consist of 2 DNA-binding sites orientated in a head-to-head manner [39, 40] separated by 15–17 nucleotides [41]. NICD dimerization is facilitated via a conserved interface in NICD's ANK (Ankyrin repeat) domain. In human N1ICD, this interface consists of Arg$^{1985}$, Lys$^{1946}$, and Glu$^{1950}$. Dimerization in NOTCH1 is effectively abolished by mutating Arg$^{1985}$ into Ala$^{1985}$ (R$^{1985}$A), Lys$^{1946}$ into Glu$^{1946}$ (K$^{1946}$E) or Glu$^{1950}$ into Lys$^{1950}$ (E$^{1950}$K), and loss of dimerization results in reduced activation of dimer-dependent targets [38, 41, 42].

Given the conservation of the dimer interface in most Notch receptors (*Caenorhabditis elegans* being the exception), the conservation of SPSs near known Notch targets [43–45], and the ability of synthetic SPSs to regulate NICD levels [46], we hypothesized that the precise mixture of agnostic and dimer-sensitive targets in a given cell will couple with Notch signal strength and dynamics to shape the responses to Notch signals and contribute to the context-specificity

of Notch-related pathologies. To test this hypothesis in a mammalian species, we evaluated mice homozygous or hemizygous for dimerization-deficient alleles of *Notch1* ($N1^{RA}$) and *Notch2* ($N2^{RA}$) and found that dimerization contributes to context-specific Notch activity. We report that in the hemizygous state, dimer-deficient Notch molecules are haploinsufficient in the heart and intestine, with lethal consequences when modified by the presence of the ecto-parasite, *Demodex musculi* (fur mites). Mice homozygous for a dimer-deficient $N1^{RA/RA}$ allele displayed a female-biased, mild cardiac phenotype consistent with Notch loss-of-function. Conversely, $N2^{RA/RA}$ mice displayed a striking over proliferation of MZBs in parasite-infested mice but not in mite-free mice. In time, the cell-type specific gain-of-function activity of $N2^{RA/RA}$ produced a splenic marginal zone lymphoma-like phenotype in mite-exposed mice. Mechanistically, these effects are consistent with a shift in target amplitude favoring mono-mer-driven targets and a potential reduction in negative feedback.

## Results

### Generation of N1 and N2 dimerization-deficient mice

To interrogate the physiological role of dimerization, we first quantified the effective binding cooperativity of NTC to SPS and CSL sites using purified RBPj, the N1 RAM-ANK domain, and a MAML protein in electrophoretic mobility shift assays (EMSAs). We compared the mobility of NTC complexes containing either wild-type (WT) N1ICD or a mutant version with the substitution of a single critical Arginine residue ($Arg^{1974}$) located in the mouse N1 ANK domain to alanine (Fig 1A and 1B, S1A–S1D Fig). Quantifying the intensity levels of the bands in the image and fitting the data to a binding model that takes into account cooperative binding (see methods), we measured the cooperativity contributed by NTC binding to either an SPS probe (1xSPS) or a CSL probe containing 2 sites (2xCSL). We find that both WT and mutant NTC had no cooperative binding to the 2xCSL probe. In contrast, when the first site was already bound, the WT NTC displayed a 5-fold stronger binding to the second site in the 1xSPS probe compared with binding to the first site. Importantly, NTC containing $N1^{RA}$ICD had no cooperative binding to the second site in the 1xSPS probe (Fig 1A, S1A–S1D Fig). Dimerization-dependent activity of N2ICD and its ability to heterodimerize with N1ICD suggests it too binds cooperatively to SPS (S1F Fig).

To ask whether loss of cooperativity impacts function in vivo, we used CRISPR-Cas9 to introduce amino acid substitutions at $Arg^{1974}$ and $Arg^{1934}$ in N1 and N2, respectively. Following the generation of a double-strand break in Exon 32 of N1 or N2, a short oligonucleotide harboring 2 linked mutations was homologously recombined. Nucleotides coding for $Arg^{1974}$ in N1 ($N1R^{1974}$) and $Arg^{1934}$ in N2 ($N2R^{1934}$) were changed to code for Ala, creating the $N1^{R1974A}$ ($N1^{RA}$) and $N2^{R1934A}$ ($N2^{RA}$) mutations, respectively (Fig 1B and 1E). To facilitate genotyping of $N^{RA}$ mice, silent mutations were introduced to abolish a BglII restriction site while creating an XbaI restriction site in $N1^{RA}$ (Fig 1B) or to generate a BglII restriction site in $N2^{RA}$ (Fig 1E). Animals that carry 1 ($N1^{+/RA}$ or $N2^{+/RA}$) or 2 ($N1^{RA/RA}$ or $N2^{RA/RA}$) mutated chromosomes were first identified by digesting PCR products of the targeted chromosomal regions in Exon 32 with XbaI and/or BglII. XbaI cuts the $N1^{RA}$ chromosome but not the WT PCR product; BglII cuts the $N2^{RA}$ but not the N2 WT PCR products (Fig 1D and 1G). The presence of each respective RA mutation in $N1^{RA}$ and $N2^{RA}$ founders was subsequently confirmed by direct DNA sequencing (Fig 1C and 1F). Animals deficient for N1 dimerization ($N1^{RA/RA}$), N2 dimerization ($N2^{RA/RA}$), or both N1 and N2 dimerization ($N1^{RA/RA}$; $N2^{RA/RA}$) were generated by crossing founders. All were viable, fertile, and without any overt phenotype at P360 in mixed or C57BL6 background (unless otherwise stated, all subsequent results were generated in the mixed background). Because Notch1 signaling plays a central role in T-cell

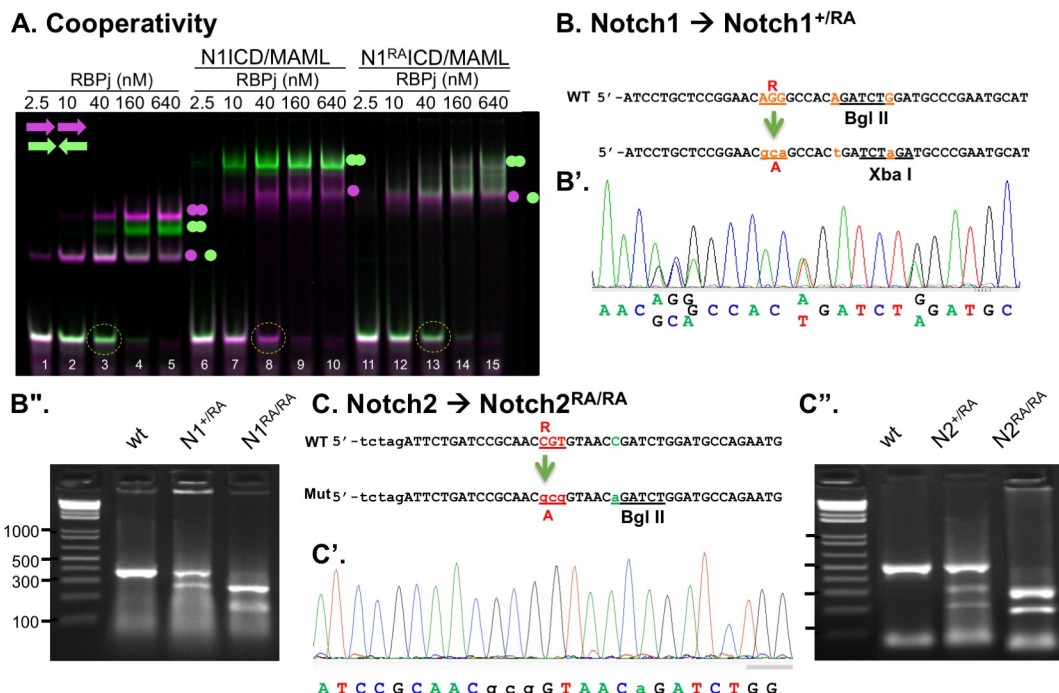

**Fig 1. Generation of Notch dimerization-deficient mice. A.** Electrophoresis mobility shift assay for purified proteins binding to CSL (magenta) or SPS (green) probes. Balls mark occupancy of 1 or 2 sites. Note that cooperative binding by WT NTC, but not the RA mutant NTC, specifically depletes the SPS probe but not the CSL probe (compare lanes 8 and 13). See text and S1 Fig for detail. **B-C.** CRISPR-Cas9-mediated double-strand break was used to mediate homologous recombination of a short oligonucleotide into Exon 32 substituting Arg $N1^{R1974}$ (**B-B"**) and $N2^{R1934}$ into Ala (**C-C"**), generating $N1^{RA/RA}$ and $N2^{RA/RA}$ animals. To facilitate genotyping, 2 silent mutations (in red) were included in the oligo, abolishing the BglII restriction site while generating an XbaI site in *Notch1* (**B'**). In Notch2, a silent mutation (in green) was included to create a BglII site (**C'**). Sequencing PCR products containing these regions confirmed the presence of the Arg to Ala substitution in founders; digestion of these PCR products with XbaI (for N1, **B"**) and BglII (for N2, **C"**) confirmed the presence of 1 (N1) or 2 (N2) mutant alleles. See S1 Data for raw data. CSL, CBF1/Suppressor of Hairless/LAG-1; MAML, mastermind-like; N1ICD, Notch1 intracellular domain; $N1^{RA/RA}$; $N2^{RA/RA}$, Notch1/Notch2 RA homozygous; NTC, Notch transcription complex; RA, Arg ($N1^{R1974}/N2^{R1934}$) to Ala substitution; RBPj, recombinant binding protein for immunoglobulin Kappa j region; SPS, sequence-paired site; WT, wild-type.

development [10], we expected that loss of Notch1 dimerization would negatively impact the T-cell compartment. However, an analysis of the T-cell compartments in the thymus and spleen as well as thymic T-cell subcompartments (CD4⁻/CD8⁻ double negative [DN]; single positive [SP], and CD4⁺/CD8⁺ double positive [DP]), revealed a normal T-cell compartment in $N1^{RA/RA}$ mice relative to WT controls (S2 Fig). These findings and the normal life span of $N1^{RA/RA}$ and $N2^{RA/RA}$ animals was surprising because we anticipated many dimerization-dependent genes (e.g., Notch-regulated ankyrin repeat protein [*Nrarp*], *Hes* [Hairy and Enhancer of Split*) 1*, *Hes5*, and *Myc*; [38, 45]) would be negatively impacted as seen when constitutively active WT and dimer mutant Notch proteins lacking the extracellular domain (NΔE and N^RA ΔE) are overexpressed in cell culture (S1E Fig).

## Homologous $N1^{RA/RA}$; $N2^{RA/RA}$ mice display barrier defects in the colon and reduced proliferation in the crypt when challenged with 1% DSS

Notch1 and Notch2 act redundantly during the development and maintenance of the gut to block Klf (Krüppel-like factor)4-induced niche exit of intestinal stem cells [47, 48] and Math1-induced differentiation of secretory cells [49, 50]. However, no pathology was revealed

in histological examination of $N1^{RA/RA}$; $N2^{RA/RA}$ intestines. Several studies have shown that a chronic decrease in Notch signaling compromised the intestinal barrier and exacerbated colitis in different mouse models [51–53]. Under the assumption that a compromised barrier may increase sensitivity to dextran sulfate sodium (DSS)-induced colitis, we asked whether intestinal homeostasis and intestinal barrier were robust in $N1^{RA/RA}$; $N2^{RA/RA}$ by exposing mice to DSS. When given 2.5% DSS in their drinking water, WT mice developed colitis, as evident by a mild weight loss, whereas co-housed $N1^{RA/RA}$; $N2^{RA/RA}$ mice exhibited severe weight loss, necessitating euthanizing within 1 week (Fig 2A). To ask if $N1^{RA/RA}$; $N2^{RA/RA}$ mice were

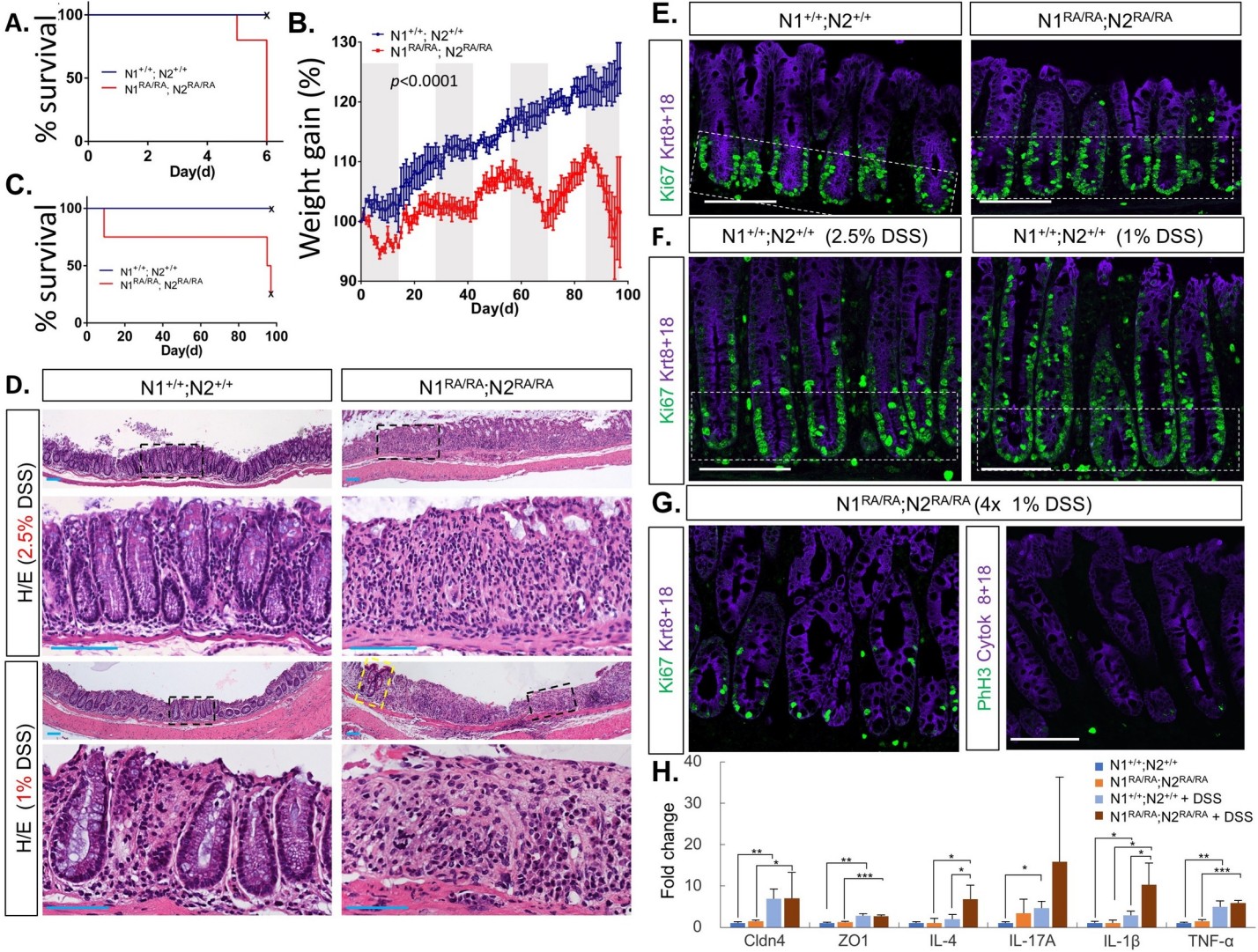

**Fig 2. Notch dimerization-deficient mice are sensitized to DSS-induced colitis. A.** All $N1^{RA/RA}$; $N2^{RA/RA}$ mice exposed to 2.5% DSS treatment had to be euthanized due to severe weight loss before day 7. **B.** Daily weight measurements of WT (blue) or mutant (red) mice treated with alternate cycles of 1% DSS (gray sections) or no DSS (white sections). **C.** Survival curve of 1% DSS-treated mice. Three of 4 mutant mice had to be euthanized due to severe weight loss by the fourth cycle. **D.** Hematoxylin-eosin staining of colonic tissue from DSS-treated mice. Dashed black boxes are enlarged below. Yellow dashed box showed a section with crypts in an otherwise injured colon in 1% DSS-treated $N1^{RA/RA}$; $N2^{RA/RA}$ mice. **E.** Ki67 staining of untreated mice with the indicated genotypes, crypt regions (Ki67+) are boxed. **F.** Increased proliferation competence in colon of DSS-treated WT mice (Krt8/18+, Ki67+ staining outside the box). **G.** Decreased proliferation competence in colon of $N1^{RA/RA}$; $N2^{RA/RA}$ mice exposed to 4 rounds of 1% DSS. (H) qPCR on RNA extracted from distal colon of $N1^{RA/RA}$; $N2^{RA/RA}$ and WT mice treated for 11 days with 1% DSS; $n = 3$. (*$p < 0.05$), S1 Data for raw data. DSS, dextran sulfate sodium; H/E, hematoxylin/eosin stain; $N1^{RA/RA}$; $N2^{RA/RA}$, Notch1 Arg[1974]Ala Notch2 Arg[1934]Ala homozygous qPCR, quantitative polymerase chain reaction; WT, wild-type.

predisposed to develop colitis, we exposed them to 1% DSS, which does not affect the weight of control littermates, even after repeated exposures (Fig 2B and 2C). By contrast, co-housed $N1^{RA/RA}$; $N2^{RA/RA}$ mice experienced significant weight loss. Some had blood in their stool at end of the third and fourth DSS cycle, and one had to be removed during the first cycle of DSS following severe weight loss and bloody stool. However, all remaining mice recovered well when DSS was removed from the drinking water, even after multiple cycles of treatment (Fig 2B). Histological analysis of the colons revealed injury in WT mice and a complete disruption of colonic epithelium of $N1^{RA/RA}$; $N2^{RA/RA}$ mice treated with 2.5% DSS (Fig 2D, top). After 4 periods of 1% DSS exposure $N1^{RA/RA}$; $N2^{RA/RA}$ mice had a more severe injury than controls or 2.5% DSS exposed WT mice (Fig 2D). Relative to baseline (Fig 2E), proliferation in WT crypt epithelia (Krt8/18 positive) was elevated in 1% or 2.5% DSS-treated controls (Fig 2F). By contrast, we noticed a surprising decrease in proliferation competence (less Ki67) and mitosis (Phospho-H3 positive cells) in colonic epithelium following the fourth 1% DSS treatment in crypts of $N1^{RA/RA}$; $N2^{RA/RA}$ mice (Fig 2G) relative to WT or untreated crypts, suggesting a role for Notch cooperativity within colonic stem cells recovering from injury. Finally, although RA-specific changes in barrier markers were not detected, an upward trend in IL17a and a significant increase in IL4 and IL1ß was observed in DSS-treated $N1^{RA/RA}$; $N2^{RA/RA}$ relative to WT (Fig 2H), consistent with enhanced inflammatory responses driving colitis in these animals. Thus, these results are consistent with $N1^{RA/RA}$; $N2^{RA/RA}$ alleles being hypomorphic loss-of-function alleles in maintaining intestinal homeostasis and immune response.

## $N1^{RA}$; $N2^{RA}$ hemizygotes display impaired intestinal cell proliferation and fate allocation in the crypt and cause ventricular septum defects in the heart

Both the WT and monomeric RA NICD proteins form NTC complexes on CSL sites as shown by EMSA (Fig 1A, S1 Fig) and chromatin immunoprecipitation (ChIP) (S4 Fig in [45]), supporting the hypothesis that the signal generated in $N1^{RA/RA}$; $N2^{RA/RA}$ mice activated sufficient non-SPS gene targets to a level needed to support Notch-dependent decisions. If the Notch RA alleles are hypomorphic, challenging RA mutant mice by lowering protein levels might reveal additional phenotypes. To test this idea, we generated an allelic series in which 1 copy of $N1$ or $N2$ was deleted in the $N1^{RA/RA}$; $N2^{+/+}$ or $N1^{+/+}$; $N2^{RA/RA}$ backgrounds, respectively, as well as deleting 1 copy of $N1$ and 1 copy of $N2$ in the $N1^{RA/RA}$; $N2^{RA/RA}$ background. Single N1 or N2 RA hemizygotes resembled RA mutants; however, crossing $N1^{RA/RA}$; $N2^{RA/RA}$ with $N1^{+/-}$; $N2^{+/-}$ animals generated significantly fewer $N1^{RA/-}$; $N2^{RA/-}$ pups at birth than expected ($p < 3\times10^{-9}$, Chi-squared distribution analysis, S1 Table).

$N1^{+/-}$; $N2^{+/-}$ double hemizygous animals have a normal life span and no overt phenotype under normal housing conditions but have been reported to develop mild cardiac phenotypes [54, 55]. Given the critical role the circulatory system plays during gestation, we suspected that the poor representation of $N1^{RA/-}$; $N2^{RA/-}$ pups at birth was due to defects in cardiac development. Analysis of E16.5 embryos born to mite-free dams revealed highly penetrant and severe ventricular septal defects (VSDs) in the hearts of $N1^{RA/-}$; $N2^{RA/-}$ embryos (Fig 3), consistent with heart defects compromising viability. Milder VSDs were also observed with lower penetrance in $N1^{RA/-}$; $N2^{+/RA}$ embryos (Fig 3). Around this time, the colony became infected with fur mites, but the infestation was not immediately detected on the sentinels. Strikingly, when we retroactively segregated the fecundity data in our colony, we observed a significant ($p = 0.02$, $\chi^2$) skewing of the gender ratio in C57BL6 $N1^{RA/RA}$ carriers, with a male:female ratio nearing 4:1 in fur mite–infested mice, as opposed to the normal 1:1 ratio in mite-free mice (S2 Table).

The longest-lived $N1^{RA/-}$; $N2^{RA/-}$ pup born to mite-infested, mixed-background dams was much smaller than its litter mates and was euthanized at postnatal day 30 (P30) when it

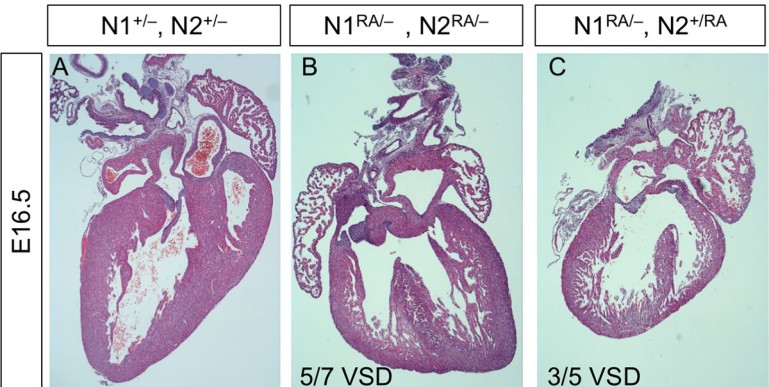

**Fig 3. Notch1$^{R1974A}$ substitution compromised ventricular septum development. A.** Normal heart with complete septum. **B.** $N1^{RA/-}; N2^{RA/-}$ heart with a severe, highly penetrant VSD. The penetrance by age shown below the image. **C.** $N1^{RA/-}; N2^{+/RA}$ heart with a milder, less penetrant VSD. E, embryonic day; $N1^{RA/-}; N2^{+/RA}$, Notch1 Arg$^{1974}$Ala Notch2 Arg$^{1934}$Ala hemizygous; VSD, ventricular septal defect.

became moribund. Necropsy revealed a striking and complete loss of the intestinal crypts and villi [48, 56–58]. Because complete failure to form an intestine would have not permitted survival outside the womb, we assumed this phenotype reflected loss of intestinal stem cells (Fig 4A and 4A'), consistent with the loss of Ki67 in DSS-challenged $N1^{RA/RA}; N2^{RA/RA}$ mice (Fig 2G). To investigate this phenotype further, we assessed proliferation and fate allocation in the developing intestine of E18-P1 animals with various allele combinations. To examine fate allocation, we stained sections with Alcian blue and analyzed the number of secretory goblet cells, which are known to expand in Notch hypomorphic backgrounds [50, 56]. The small intestines of control or $N1^{+/-}; N2^{+/-}$ P0 pups were indistinguishable (Fig 4B and 4C). By contrast, Alcian blue staining revealed a significant expansion in the goblet cell compartment at P0 in $N1^{RA/-}; N2^{RA/-}$ pups born to fur mite–infested dams (Fig 4D). In mice free of mites, the overall morphology appeared normal with few regions displaying excess goblets. To assess proliferation, we stained adjacent intestinal sections with antibodies against Ki67 and Phospho-H3. Strikingly, whereas trans-hemizygote $N1^{+/-}; N2^{+/-}$ intestines resembled WT (Fig 4E and 4F), the $N1^{RA/-}; N2^{RA/-}$ intestine from mite-infested dames contained very few Ki67 positive cells (Fig 4G and 4H). We also noted a reduction in the number of cells undergoing DNA replication in particular, and competence to self-renew in general, in $N1^{+/-}; N2^{RA/RA}$ intestines (S3C, S3C', S3G and S3G' Fig). However, upon eradication of *D. musculi* with permethrin, all E18.5 and all but 1 of 8 surviving P0 $N1^{RA/-}; N2^{RA/-}$ pups had at least some regions within the small intestine in which Ki67-positive tissue could be observed (Fig 4J, the percentage of Ki67+ crypts; S3I–S3J' Fig; note reduced proliferation in Ki67-positive crypts of a P1 pup post-mite eradication but coinciding with microbiome colonization). Overall, these data reflect an impaired ability to maintain intestinal stem cells in $N1^{RA/-}; N2^{RA/-}$ mice, which was exacerbated by *D. musculi* infestation. Because these phenotypes can reflect complex interactions between multiple tissues in the $N1^{RA/RA}; N2^{RA/RA}$ and $N1^{RA/-}; N2^{RA/-}$ mice, we deferred the mechanistic analysis of this gene-environment interaction in intestinal stem cells (ISC) maintenance to future investigation.

## The $N2^{RA}$ allele drives expansion of MZBs in *D. musculi*–infested mice

MZBs are exquisitely sensitive to $N2$ dosage [13–17, 19, 20]. Even a mild reduction in N2ICD can generate a noticeable decline in MZB cell numbers [20]. We used fluorescence-activated cell sorting (FACS) to analyze the splenic B-cell population and asked if $N2^{RA}$ displayed

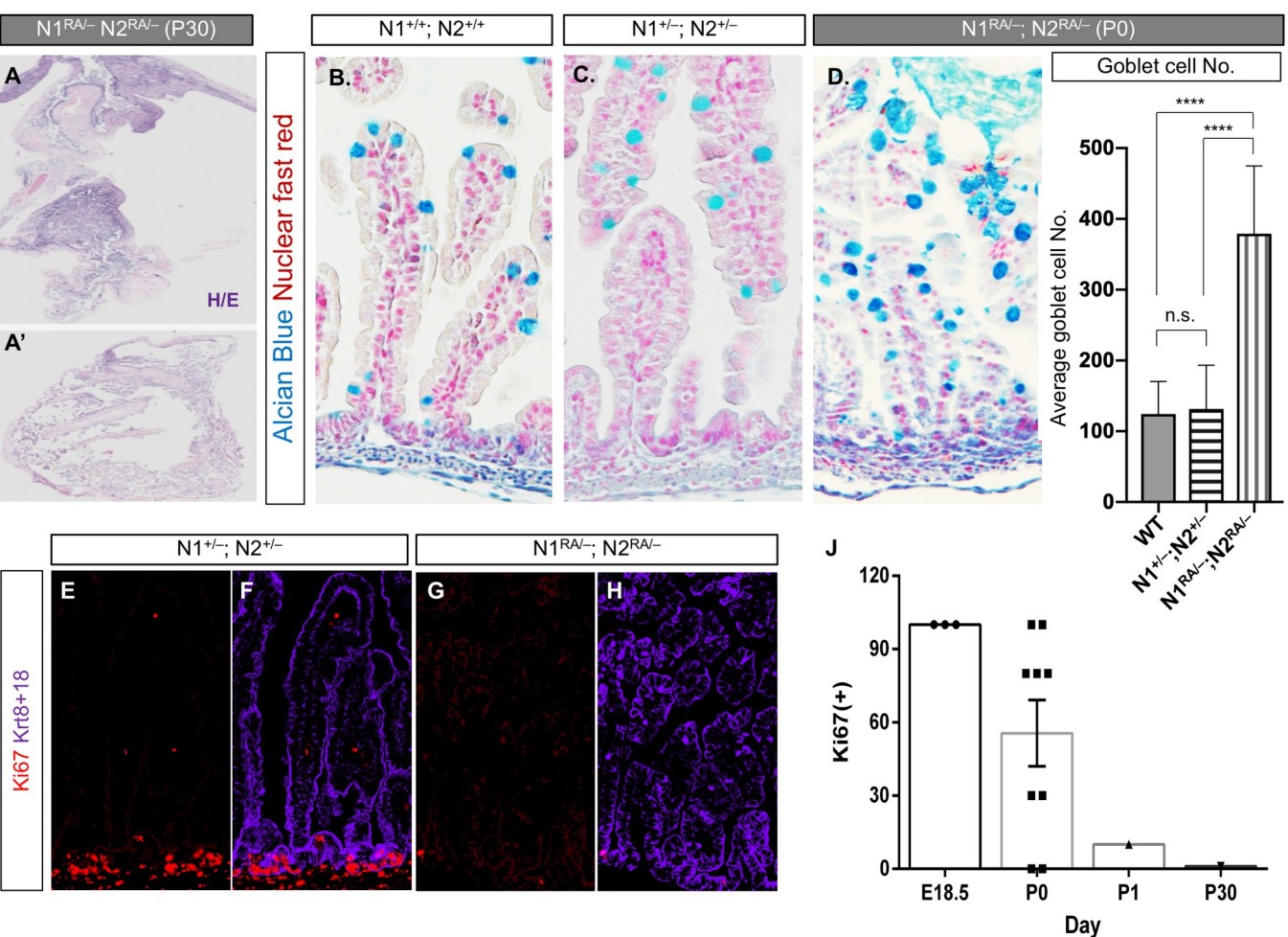

**Fig 4. Notch dimerization-deficient mice are hypomorphic for Notch activity in the gut. A.** Deletion of one N1 and one N2 allele in the RA background ($N1^{RA/-}$; $N2^{RA/-}$) caused lethality; loss of the intestine (presumably due to stem cell exhaustion) was evident in the longest surviving pup at P30. **B-C.** Alcian blue staining of $N1^{RA/RA}$; $N2^{RA/RA}$ and $N1^{+/-}$; $N2^{+/-}$ adult intestines. **D.** Alcian blue analysis of P0 intestines detected an increase in goblet cell numbers in $N1^{RA/-}$; $N2^{RA/-}$ (E-H) Ki67 staining in P0 intestine from control $N1^{+/-}$; $N2^{+/-}$ (**E, F**) or $N1^{RA/-}$; $N2^{RA/-}$ (**G, H**) newborn. **J.** Quantification of Ki67-positive area at the indicated age, among surviving $N1^{RA/-}$; $N2^{RA/-}$ pups during ($n = 2$) and after ($n = 12$) fur mite infestation; raw data in S1 Data. E, embryonic day; H/E, hematoxylin/eosin stain; $N1^{RA/-}$; $N2^{RA/-}$, Notch1 Arg$^{1974}$Ala Notch2 Arg$^{1934}$Ala hemizygous; P, postnatal day; WT, wild-type.

haploinsufficient phenotypes in the dose-sensitive MZ compartment. Intriguingly, we observed a significant increase in the MZB compartment in $N2^{RA/RA}$ spleens in a mixed-background mouse exposed to *D. musculi* (Fig 5A). The size of other splenic B-cell subsets, including the MZB precursors (MZPs), follicular B-cells (FoB) and transitional type-2 cells (T$_2$) did not change significantly in either $N2^{RA/RA}$ (Fig 5A), $N2^{+/-}$ or $N2^{RA/-}$ (Fig 5B), inconsistent with a fate switch.

MZB cells reside in the splenic marginal zone where they surveil for blood-borne pathogens. Upon encountering antigens, they rapidly differentiate into immunoglobulin M (IgM)-producing plasmablasts that secrete vast amounts of IgM before undergoing apoptosis [59]. To test IgM levels in vivo, we performed ELISA on serum collected from unstimulated $N2^{RA/RA}$ and littermate controls and detected no significant difference in the levels of circulating IgM (Fig 5C). To test for the possibility that loss of N2 dimerization negatively impacted differentiation into IgM-producing plasmablasts upon stimulation, we FACS isolated MZB and FoB cells from mite-exposed $N2^{RA/RA}$ and WT spleens. We then stimulated these cells in vitro with

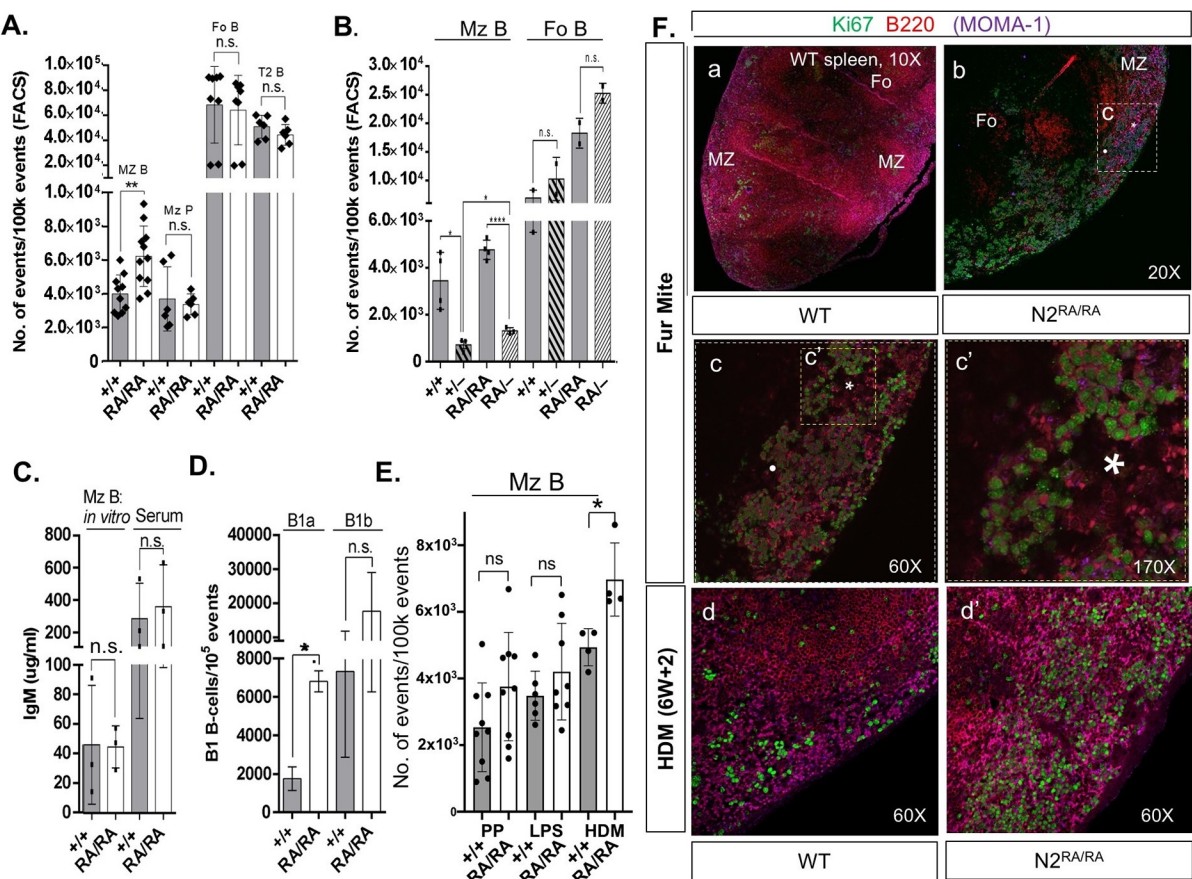

**Fig 5. Loss of Notch2 dimerization expands the splenic MZB compartment in the presence of mites. A.** FACS analysis of $N2^{RA/RA}$ fur mite–infested mouse spleens revealed an expansion the MZB compartment relative to WT (+/+). Values for MZP, F0, and $T_2$ splenic B-cell populations were not significantly different; $n = 11$. **B.** The MZB cell compartment in heterozygotes. **C.** LPS-induced differentiation and IgM production in vitro and serum levels of IgM in vivo; $n = 4$. **D.** Peritoneal B1a B-cell compartment in mite-infested WT and $N2^{RA/RA}$ mice ($n = 2$). **E.** The MZB cell compartment in $N2^{RA/RA}$ mice after treatment with permethrin, an immunosuppressant used to manage fur mite infestation (PP), after LPS, or after 6 weeks of treatment with HDM extract. ($^*p < 0.05$). Note, although a slight trend is seen in mite-free animals, power analysis indicates it would take 212 mice displaying this trend to reach significance. For LPS-treated mice, it would take 41 animals displaying this trend to reach significance. Only 4 mice were sufficient to reach significance after treatment with HDM. In (**A-E**) $p$-values generated by 2-tailed $t$-test. Error bars represent SD. **F.** Proliferation in the splenic marginal zone of mite-infested mice: (Fa) The spleen from a mite-infested control animal or (Fb) from $N2^{RA/RA}$ mice. Boxed region magnified in Fc and again, in Fc', asterisk and dot provided for orientation. Magnification noted. (Fd, d') 60× view of marginal zone from control and HDM-treated $N2^{RA/RA}$ mice. Raw data in S1 Data. F0, Filial generation 0; FACS, fluorescence-activated cell sorting; FoB, follicular B-cell; HDM, house dust mite; IgM, immunoglobulin M; LPS, lipopolysaccharide; MZB, marginal zone B-cell; $N2^{RA/RA}$, Notch 2 Arg$^{1934}$Ala homozygous; ns, not significant; P, postnatal day; PP, post-permethrin; $T_2$, type 2 cell; WT, wild-type.

lipopolysaccharide (LPS) for 5 days followed by imaging to assess for proliferation and by ELISA to quantify the IgM levels secreted into the medium. Relative to WT MZB cells, there was no appreciable difference in the ability of $N2^{RA/RA}$ MZB cells to respond to LPS stimulation (S4A Fig), and ELISA analysis detected no significant difference in the amount of IgM secreted by $N2^{RA/RA}$ MZB cells relative to WT controls (Fig 5C). Finally, we tested whether loss of dimerization affected the development of B1 B cells, which predominantly reside in the peritoneal cavity [16]. Together, these data suggest that MZB cell numbers did not increase because of a fate switch or a failure to differentiate properly and are consistent with a proliferative phenotype triggered by the $N2^{RA}$ mutation. Notably, FACS analysis also revealed a significant expansion of B1a subset of B1 B-cells between $N2^{RA/RA}$ and WT animals but no changes in the levels of the B1b subset (Fig 5D).

In the course of these experiments, the *D. musculi* infestation was eradicated with permethrin. Within a few months, as we saw with other phenotypes, mite-free $N2^{RA/RA}$ animals no longer showed significant MZB expansion (Fig 5E; post-permethrin [PP]). Because permethrin has been reported to suppress the immune system in mice [60, 61], we assumed permethrin transiently suppressed MZB proliferation. Multiple subsequent analyses over many months, however, failed to detect a significant difference in MZB numbers between permethrin-treated $N2^{RA/RA}$ and control mice post eradication, suggesting that this allele was indistinguishable from WT in untreated and pathogen-free animals. Accordingly, we compared MZB numbers between $N2^{+/-}$ spleens (which have decreased MZB numbers due to haploinsufficiency) with $N2^{RA/-}$ spleens and found no further reduction in MZB numbers [13, 20]. These data suggest that the $N2^{RA}$ allele has neither a haploinsufficient character for the $N2^{RA}$ allele in the MZB lineage (Fig 5B) nor a proliferative phenotype triggered cell-autonomously by the N2$^{RA}$ protein. To ask whether environmental stimuli interacted with N2$^{RA}$ within the spleen, mice were exposed to bacterial lipopolysaccharide (LPS) or to dermatitis, produced by twice weekly exposure of the ear epidermis to house dust mite extract (HDM, *Dermatophagoides farinae*) for 6 weeks (see Methods). MZB numbers were analyzed 8 weeks after the first LPS injection or HDM extract application. LPS-induced transient IgM production but did not significantly change MZB census in the spleen at 8 weeks (Fig 5E, note a trend towards increase was observed in the $N2^{RA}$ LPS-treated mice). Importantly, HDM-induced dermatitis elevated MZB numbers only in the $N2^{RA/RA}$ mice but not in controls (Fig 5E, HDM). Collectively, these experiments establish that specific gene-environment interactions produce the MZB phenotype in $N2^{RA/RA}$ mice.

To assess the rate of splenic B-cell proliferation in mite-infested mice, we stained C57BL/6J $N2^{RA/RA}$ and WT (WT) spleen sections with antibodies against Ki67, B220 (to mark the B-cells), and Moma-1 (to mark the marginal zone). We detected defuse B220 staining with elevated Ki67 in marginal zone cells in the $N2^{RA/RA}$ spleen (Fig 5F), as well as in germinal centers (rich in FoB cells and strongly B220 positive, S4 Fig), not seen in WT spleen of co-housed animals (S4B–S4D Fig). Increased marginal zone proliferation in $N2^{RA/RA}$ relative to WT controls was also detected in spleens after 6 weeks of HDM treatment (Fig 5F, 5D and 5D'). Combined, these data suggest that mite infestation or prolonged HDM exposure lead to enhanced MZB proliferation.

## Aged $N2^{RA/RA}$ mice have enlarged spleens but only *D. musculi*–infested mice progress to a splenic MZB lymphoma–like state

As noted here, we observed no change in life span in $N2^{RA/RA}$ or $N1^{RA/RA}$; $N2^{RA/RA}$ animals. However, as old animals were culled, we noticed enlarged spleens in animals carrying the $N2^{RA/RA}$ allele post mite eradication (S5A Fig). Spleens from mice infested by *D. musculi* were much larger, the most severe splenomegaly exhibited by $N2^{RA/RA}$; $N1^{+/RA}$ (Fig 6A–6E), perhaps because they contained a higher contribution of the inflammation-prone Friend virus B–susceptible (FVB) strain [62, 63]. Histologically, spleens from mite-infested $N2^{RA/RA}$ carriers lost their typical architectures with $N2^{RA/RA}$; $N1^{+/RA}$ spleens showing the greatest disruption. Splenic morphology showed an expansion of the white pulp at the expense of the red pulp. In the most severe cases, the spleens appeared to consist of white pulp only (Fig 6F–6K). In addition, 2 $N2^{RA/RA}$; $N1^{+/RA}$ animals had multiple enlarged abdominal lymph nodes. H/E staining revealed a striking morphological similarity to the enlarged spleens from the same animals, suggesting infiltration by a splenic population, or lymphoproliferation within the LN (Fig 6L–6L"). To further characterize the nodes, we stained the enlarged spleens and lymph nodes for the B-cell markers B220 and Pax5. Both were positive for the B-cell markers (Fig 6M–6N').

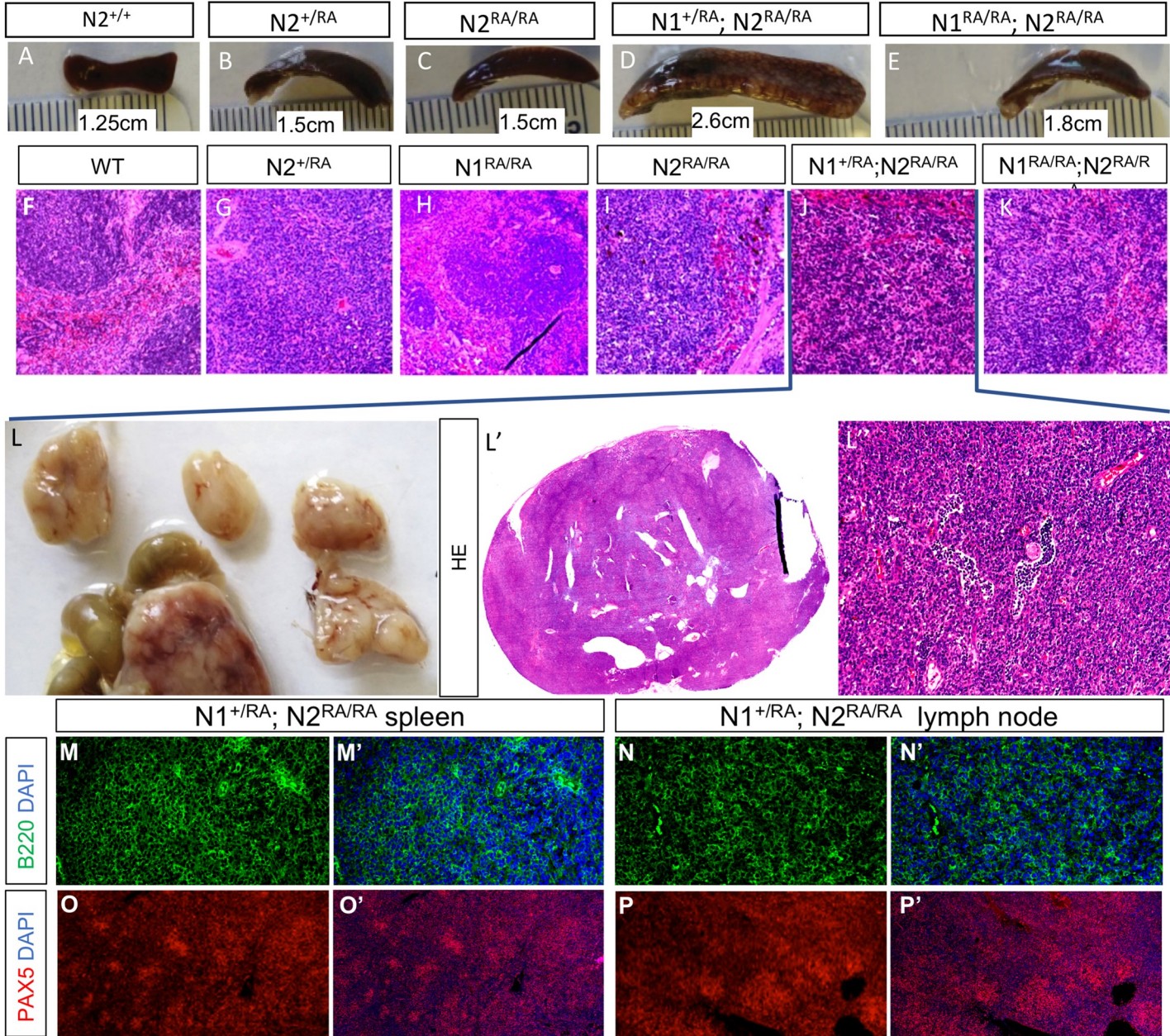

**Fig 6. Aged *N2^{RA/RA}* and *N2^{RA/RA; N1+/RA}* mice develop severe splenomegaly and tumors reminiscent of SMZL. A-E.** Images of spleens from mice P600-P700 with indicated genotypes. **F-K.** Severely enlarged spleens are associated with disorganized morphology and the appearance of cells with large nuclear:cytoplasmic ratio (genotypes indicated, see low magnification image in S6E Fig). **L.** Large enteric lymph nodes detected in the individual shown in (**J.**), densely populated with cells with large nuclear:cytoplasmic ratio (**L'-L".**). Staining the spleen (**M.**) and lymph node (**N.**) with Ki67 and the B-cell markers B220 and Pax5 identifies infiltrating cells as spleen-derived B cells. Magnification: 10×. *N2^{RA/RA; N1+/RA}*, Notch1/Notch2 RA homozygous; RA, Arg (*N1^{R1974}/N2^{R1934}*) to Ala substitution; SMZL, splenic marginal zone B-cell lymphoma.

These features are reminiscent of human splenic MZB lymphoma (SMZL), a clonal B-cell neoplasm that manifests itself in elderly patients and consists of small lymphocytes that infiltrate the lymph nodes and other organs [23–25]. As would be expected, the enlarged lymph nodes and spleens were highly proliferative as indicated by Ki67 and phospho-H3 staining (S5B and S5D Fig). Paradoxically, human SMZL has been associated with hyperactive NOTCH2 signaling due to an N2ICD-stabilizing truncation upstream to the WSSSP sequence in the Pro-Glu-

Ser-Thr–rich (PEST) domain of NOTCH2 [64–66], whereas the $N2^{RA}$ alleles are hypomorphic in expression assays and in the context of the heart, crypt, and gut barrier. To confirm that this line of CRISPR-modified mice did not inherit a truncated *Notch2* allele, we sequenced Exon 34 DNA isolated from lymph node filled with B-cells and found a perfect match to the WT *Notch2* allele (ENSMUSG00000027878).

Recently, we identified an unanticipated consequence of increasing the number of SPS sites in the *Drosophila* genome, namely, accelerated degradation of phospho-NICD [46]. Further, we demonstrated that this enhanced degradation affects some, but not all, Notch-dependent decisions in *Drosophila*. Bristle precursor cells requiring a pulse of Notch were refractory to the destabilizing effect, whereas wing margin cells reliant on prolonged Notch signals were sensitive to NICD degradation [46]. To test whether the similarity of our phenotype to SMZL might in part be reflective of N2ICD stabilization, we assessed N2ICD levels in nuclear extracts from WT and $N2^{RA/RA}$ MZB cells. Although we see high variability between replicates, we find no evidence for a significant accumulation of nuclear N2ICD in sorted $N2^{RA/RA}$ MZBs (S6 Fig). These findings suggest that the gain-of-function phenotypes observed in $N2^{RA/RA}$ MZB cells are not due to enhanced N2ICD stabilization. Thus, unlike stabilization seen in many human SMZL patients, increased N2ICD stability is unlikely to be the cause of the SMZL phenotype in the $N2^{RA/RA}$ mice.

## Loss of Notch dimerization does not impact chromatin accessibility

In $N2^{RA/RA}$ mice, Notch pathway activity in T cells and skin is intact, as it relies on *Notch1*. Assuming the MZB effect is cell autonomous, we asked if the mechanism involved in enabling a haploinsufficient allele to drive MZB expansion involves changes in chromatin accessibility. We performed ATAC-Seq on MZB cells isolated from WT and $N2^{RA/RA}$ mice in the absence of fur mites (4 biological replicates each). We mapped reads under the peaks and used EdgeR (https://bioconductor.org/packages/release/bioc/html/edgeR.html) to identify differential accessibility across the genome between samples (see Methods for details). A total of 89,059 peaks were mapped, of which 87,575 were present in both $N2^{RA/RA}$ and WT MZB cells with all 8 samples being highly related ($R > 92\%$; Spearman Correlation; Fig 7A). Hence, genome accessibility changed minimally between these MZB cells with only 984 peaks (1.1%) enriched in $N2^{RA/RA}$ and 500 (0.56%) in wild type. We next used GREAT to assign genes to all peaks. Based on these assignments, only 108 were unique to WT, and 122 were unique to $N2^{RA/RA}$. Importantly, none of the differently enriched peaks were present at the *Myb* or *FoxM1* loci, which are known to drive of B-cell proliferation [67] or next to a known Notch target *Dtx1* [13] (Fig 7B). Next, we analyzed DNAse hypersensitive peak upstream of the *Myb* locus in the ENCODE datasets and identified an enhancer accessible in several cell types, including the kidney. The genome of a kidney-derived cell line used to overexpress constitutively active NΔE and a dimer-deficient version, NΔERA (S1 Fig) was examined by SplitDamID [45]. We detected strong binding to the *Myb* enhancer using Notch/RBP complementing pairs but not by Notch dimers. These data suggest that the *Myb* enhancer can respond to monomeric NTC and is insensitive to loss of dimeric NTC activity. Collectively, these data suggest that the changes in MZB gene expression in the $N2^{RA/RA}$ animals are not due to changes in chromatin accessibility but may reflect changes in gene expression caused by the loss of Notch dimer NTC complexes at some other loci, most likely the SPS-dependent negative regulators *Hes1*, *Hes5*, and *Nrarp* [45, 68].

## Discussion

The possibility that DNA-binding site architecture contributes to Notch signaling outcomes has been considered in past studies [43, 44]. NTCs can bind to enhancers with CSL binding sites as monomers that function independently of one another. In such a model, the

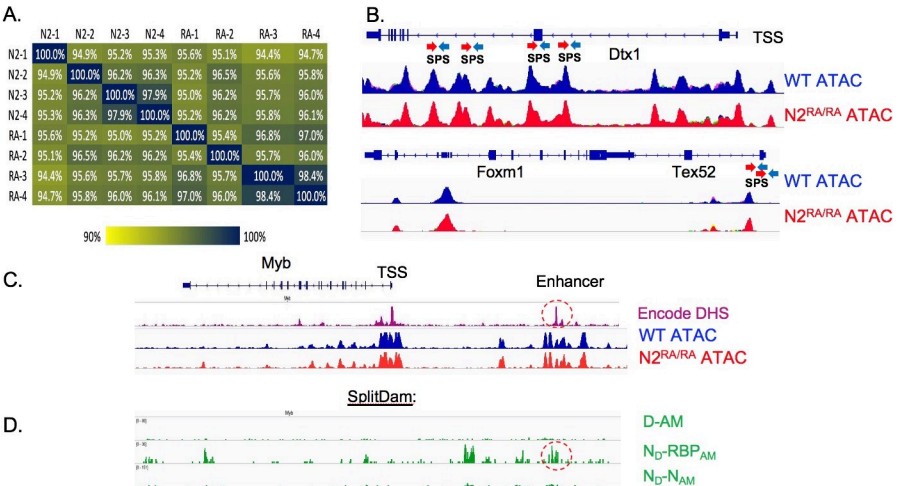

**Fig 7. Mite-infested $N2^{RA/RA}$ activate a proliferation module in MZB. A.** Spearman correlation table measures ATAC-Seq peaks similarity matrix between 8 mice (4 in each genotype). All are >92% similar. **B-C.** ATAC-Seq data reveal that DNA accessibility was not altered near the Notch target *Dtx1* that is down-regulated in expression in $N2^{RA/RA}$ animals (**B.**). WT in blue, $N2^{RA/RA}$ in red. Location of validated enhancers containing SPSs are shown. Note, each dimer-dependent site can be composed of either 2 canonical sites (red arrows) or 1 canonical and 1 noncanonical site (blue arrow). Close inspection of ATAC peaks and the proliferation drivers *FoxM1i* (**B.**) or *Myb* (**C.**) loci. **D.** The *Myb* locus was analyzed for chromatin accessibility (Encode DHS and ATAC-seq data generated in this study) and DNA methylation by DAM methyl transferase complementation (SplitDAM). The methylation patterns generated by control D/AM halves are compared to methylation patterns generated by Notch-D/RBPj-AM pairs (which recognizes both dimer-dependent and dimer-independent sites) and Notch-D/Notch-AM pairs (which recognizes only dimer-dependent sites). See text and [45] for additional details. ATAC-Seq, Assay for Transposase Accessible Chromatin sequencing; D/AM, complementing halves of DAM; DAM, DNA adenine methyltransferase; DHS, DNAse hyper sensitive; MZB, marginal zone B-cell; $N2^{RA/RA}$, Notch-D, Notch1 fused to the D half of DAM; *Notch2* Arg $N2^{R1934}$ to Ala substitution, homozygous; RBPj-AM, recombinant binding protein for immunoglobulin Kappa j region fused to AM half of DAM; SPS, sequence-paired site; TSS, transcription start site; WT, wild-type.

probability of target activation will depend upon the relative abundance of nuclear NICD and the number of CSL sites at a given enhancer [69–71] and/or burst size [72, 73]. SPSs are found in enhancer regions of up to 30% of mammalian Notch targets including *Nrarp/NRARP*, *Hes1/HES1*, *Hes5*, and *Myc* [30, 38, 42, 74] We have previously documented that approximately 2,500 SPSs were bound by N1ICD dimers in the mouse genome and that dimerization-deficient Notch molecules bind poorly to SPSs in vivo, even at high NICD concentrations [45]. In the murine kidney cell line, mK4, about 15% of Notch targets required NICD dimerization. SPSs and cooperativity also proved critical for the oncogenic activity of N1ICD in murine T cells, where NICD dimerization at a distant enhancer was required for *Myc* activation [42]. Thus, Notch targets are either agnostic to dimerization or sensitive to its presence, leading to the hypothesis that cooperativity may modulate Notch responses at physiological NICD concentrations. Recently, evidence that SPSs contribute to an increased probability of transcriptional activation and increased transcriptional burst duration during *Drosophila* embryonic development was reported [72].

Although our study concludes that embryonic development in the mouse proceeds normally in the presence of Notch receptors lacking cooperativity, it nonetheless uncovered important contributions of cooperativity to mammalian development and homeostasis under environmental stress. Notch$^{RA}$ phenotypes are greatly enhanced by reduced gene dosage in *Notch1*, which causes VSDs ([75–77]) or in both *Notch1* and *Notch2* in the gut. The developmental phenotype uncovered when $N1^{RA}$ dose is reduced (in $N1^{RA/-}$) resembles other Notch pathway deficiencies known to cause a wide range of heart defects, including VSDs, and are

milder than a weak hypomorphic Notch1/2 allele on the B6 background [20]. Interestingly, dosage alone did not impact the ability of the $N2^{RA}$ allele to control development and homeostasis of the dosage-sensitive MZBs in parasite-free mice, even in the hemizygote state ($N2^{RA/-}$).

Strikingly, loss of cooperativity-dependent contributions to homeostasis becomes acute when mutant animals are burdened with exoparasites, namely, fur mites. Interestingly, when the environment includes unmanaged exoparasite infestation, dimerization-deficient cooperativity mutant receptors can behave as hypomorphic (loss-of-function) alleles in some contexts (intestinal barrier formation, ventricular septum formation, intestinal stem cell self-renewal, female survival) yet remain indistinguishable from the WT allele in many other tissues or can trigger a similar disease as a gain-of-function allele (SMZL) in a cell type that otherwise appeared unaffected by the mutation. Note that in $N2^{RA/RA}$ mice, Notch pathway activity in T cells and skin, where *Notch1* is present, are identical to the WT. Thus, the defect is more likely to reside in the B-cell lineage than elsewhere, although this remains to be examined more fully in the future. We propose these behaviors can be explained by assuming that the balance between targets agnostic to dimerization (like *Myb*) and those dependent on cooperativity on SPSs (like *Hes1*, *Hes5*, *Nrarp*) varies by cell type (Fig 8). Moreover, these differences in dependence upon SPS versus non-SPS target gene regulation between tissues can be further exposed by additional stressors such as environmental insults (i.e., exoparasites) or changes in gene dose. We propose that accessible SPSs can act as a "sink," holding on to NICDs that otherwise will be available to regulate monomer-dependent enhancers. If true, more $N2^{RA}$ICD may be available to regulate key targets. Importantly, this insight may translate to other transcription factors where variants of uncertain significance are associated with developmental syndromes or neoplastic disease. Such mutations may control molecular behaviors integrating the environment with specific cooperative interactions in affected tissues.

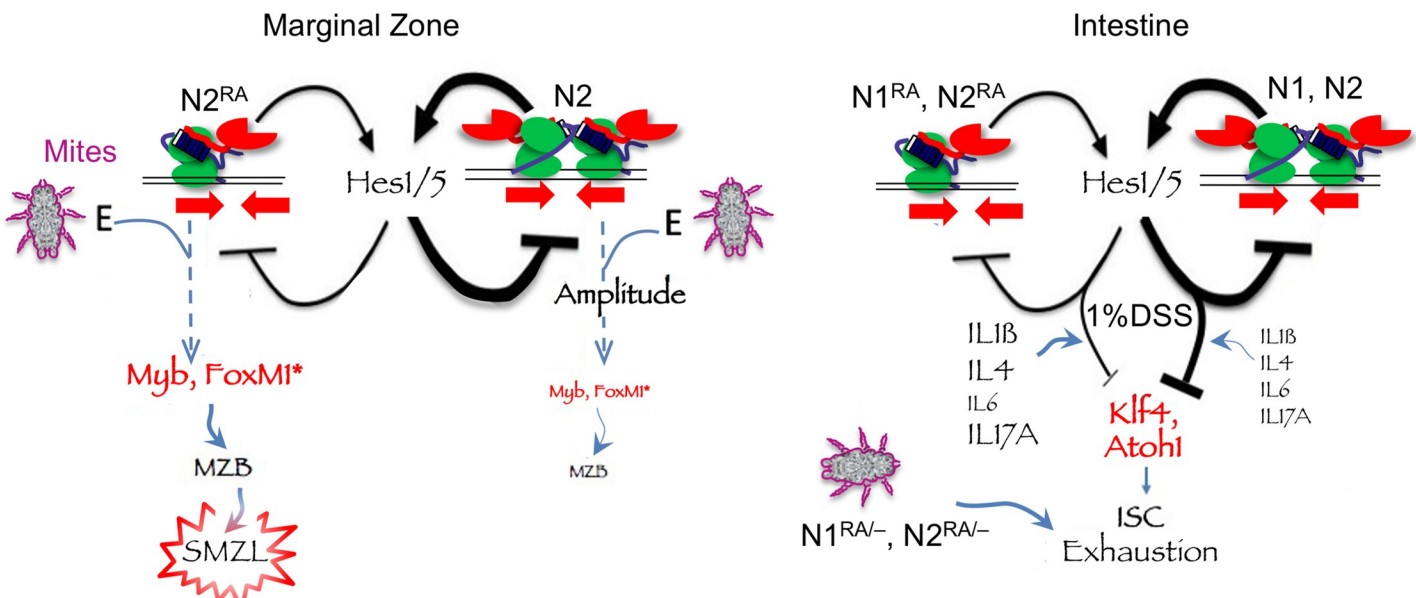

**Fig 8. A schematic summary of the findings with a hypothetical mechanism.** *Notch* integrates environmental cues (mites, cytokines) to drive proliferation in MZB (*Notch2*) and block differentiation in (ISCs, *Notch1* and 2). Activation of SPS-dependent *Hes* repressors creates a negative feedback tuning of transcription amplitude in MZB and driving the response magnitude in ISCs. In *Notch*^RA/RA^ mutants, SPS-dependent *Hes* gene expression is dampened, and increased availability of NICD^RA^ to monomer-driven target due to reduce trapping on accessible SPS may also contribute. Repeated DSS treatment, or hemizygosity in *Notch*^RA/RA^ mutants, exposed insufficient blockade of pro-differentiation signals in ISC, most likely delivered via immune cells or cytokines (IL1ß, 4, or 17A) in DSS-treated mice. Fur mite infestation creates an environment for runaway proliferation in MZB, and enhanced dosage effects in *Notch*^RA/RA^ mutants leading to complete loss of ISC postnatally even without DSS challenge. Asterisk marks hypothetical targets selected for their known function in driving B-cell proliferation [67]. DSS, dextran sulfate sodium; ISC, intestinal stem cell; MZB, marginal zone B-cell; NICD, Notch intracellular domain; SMZL, splenic marginal zone B-cell lymphoma; SPS, sequence-paired site.

The diverse cell types responding to a cutaneous parasite must reflect a systemic change driven by the parasite. Because we could replicate the impact on MZB proliferation in short exposures to dust mite extract, live parasite is not needed, and the response is likely mediated by an immune cell (or cytokine) intermediate. Dermatitis is typically associated with both skin-produced cytokines (e.g., IL-33, -17C, and TSLP), and a skewing of CD4+ T cells into T helper 2 (Th2) and their associated cytokines (e.g., IL-4,-5,-13, [78]). Similarly, fur mite infestation has been shown to increase the level of different cytokines (e.g., IL-6 and IL-12; [79] and IL-4 and -5 [80, 81]). Although the exact axis linking anti-parasite immune responses and splenic marginal zone lymphoma, intestinal homeostasis, female/male ratios, and heart development are yet to be identified, this study highlights the role of the environment in chronic disease, including ulcerative colitis and a smoldering malignancy such as SMZL. Combined with the significantly better outcomes in SMZL patients with activated *Notch2* alleles [65], one has to wonder about the role of the immune environment in onset and progression of human SMZL patients with *Notch2* mutations.

## Methods

### Ethics statement

Mice were housed at CCHMC's animal facility in accordance with IACUC's animal welfare regulations. The use of animals for the experiments described in this work was approved through protocol number IACUC2018-0105.

### Animals

To generate the RA alleles, 3 gRNAs (g25, g26, and g27) that target sites surrounding the $R^{1934}$ codon in N2 were tested for their genome editing activities. Briefly, plasmids carrying the guide sequence under a U6 promoter and Cas9 were transfected into MK4 cells. Cells were harvested 2 days after transfection and PCR amplification, with primers flanking the target sites, was performed on genomic DNA isolated from the transfected cells. PCR products were then subjected to T7 endonuclease I (T7E1) assay. Cleavage products generated by T7E1 are indicative of indels present in the PCR products as a result of genomic DNA breakage caused by Cas9-gRNA and the subsequent repair by NHEJ. A gRNA (g3) for Tet2 that was known to have good editing activity was included as positive control. Sixteen mice were obtained from injection of g27 (GGCATCCAGATCGGTTACA), Cas9 mRNA, and a donor oligo (shown in Fig 1, g27 underlined in blue) into 1-cell embryos. Digestion of PCR products with BglII showed that 8 contained products with the BglII site, 4 of which were homozygotes. Sequencing of PCR products confirmed that all had the correct $R^{1934}$A mutations. Because we mated the homozygote mice to each other, we identified potential second site mutations with the CCTop tool ([82]; https://crispr.cos.uni-heidelberg.de). g27 matches to Notch2 and to a few loci with 4 mismatches (*Lemd1*, *Ccm2*, *Papolg*, *Tubb2b* and *Tubb6*) falling within an Exon and next to a PAM sequence (no loci had fewer than 4 mismatches to this gRNA). PCR products from potential off-target sites were sequenced, and no mutations were identified in DNA isolated from any of the founders. For Notch1, a similar process led to the selection of a gRNA (CATTCGGGCATCCAGATCTG), which was injected with Cas9 mRNA and a donor oligo carrying mutations that change $R^{1974}$ to alanine into 1-cell embryos. PCR products from 24 animals were digested with BglII and XbaI, respectively. Only 1 heterozygote founder carried the $R^{1974}$A mutation, confirmed by sequencing. This founder was outcrossed into our mixed colony, having a single possible 4-mismatch target in the pseudogene *Vmn1r-ps6*. In the C57BL6line, the same gRNAs were used. A single heterozygote founder pair was bred within the C57BL6line.

## DSS treatment/colitis induction

For colitis induction, mice were given 1% to 2.5% DSS in autoclaved drinking water for up to 14 days. Weight was measured every day, and stool was checked for changes to monitor disease progression; in case of severe weight loss (>15%) or blood in stool, mice were euthanized. For recovery, mice were given normal drinking water for 14 days before starting the next cycle of DSS treatment, up to 4 cycles of injury and recovery. At the end of treatment, mice were euthanized, and tissue was collected for analysis.

## LPS injection/dermatitis induction

To stimulate the immune system with bacterial antigens, we injected adult mice intraperitoneally twice with LPS (0.5 ug/kg bodyweight, once a month, Table 1) over an 8-week period. Blood was collected from the submandibular vein once per week to analyze IgM production by ELISA. The mice were euthanized after the eighth week and their spleens were harvested for histological analyses.

Mild dermatitis was induced by application of HDM (*Dermatophagoides farina*, Table 1) extract on the ears as described [83]. In brief, the ear lobes of adult mice were gently stripped 5 times with surgical tape and 20 ug of HDM dissolved in autoclaved PBS was administered to each ear twice weekly for 6 weeks. Mice were euthanized, and spleen harvested for histology and MZB numbers determination.

## Antibodies

A full list of antibodies is provided in the Key Resource table (Table 1). The following antibodies were used for FACS analysis: Anti-mouse/human CD45R/B220, anti-CD23, anti-CD21/CD35 (CR2/CR1), anti-CD93 [AA4.1], anti-mouse IgM, anti-IgD, anti-CD45.1, anti-CD45.2, anti-CD3ε, anti-Ly-6G/Ly-6C(Gr-1), anti-CD11b, anti-TCRb, and anti-CD5 were all purchased from Biolegend. Anti-Mouse CD19 was purchased from BD Biosciences. Rabbit Anti-Ki67 (Novocastra), rabbit anti-cleaved caspase 3 (Cell signaling), rabbit anti-Phospho-Histone H3 (Ser10) (Cell signaling), rat anti-Pax5 (Biolegend), Guinea pig anti-Cytokeratin 8+18 antibody, and FITC rat anti-mouse/human CD45R/B220 (BioLegend) were used for fluorescence immunohistochemistry. Anti-rabbit Alexa Fluor Cy3 (Jackson ImmunoResearch), anti-rabbit Alexa Fluor 647 (Jackson Immunoresearch), anti-rat Cy3 (Jackson Immunoresearch), and Alexa Fluor anti-guinea pig 647 secondary antibodies were used for in fluorescence immunohistochemistry. Notch2 (D76A6) XP (Cell signaling) and HRP-linked rabbit IgG (GE healthcare) secondary antibodies were used for western blot analysis.

## Protein purification and EMSA

The mouse RBPj (aa 53–474), mouse N1ICD (aa 1744–2113), mouse N1$^{RA}$ICD (aa 1744–2113), and human SMT3-MAML1 (aa 1–280) proteins were expressed and purified from bacteria using affinity (Ni-NTA or Glutathione), ion exchange, and/or size exclusion chromatography as previously described [84]. The purity of proteins was confirmed by SDS-PAGE with Coomassie blue staining, and concentrations were determined by absorbance measurements at UV280 with calculated extinction coefficients. EMSAs were performed using native polyacrylamide gel electrophoresis as previously described [85, 86]. Fluorescent labeled probes (1.75 nM/each probe) were mixed with increasing concentrations of purified RBPj protein in 4-fold steps (from 2.5 to 640 nM) with or without the indicated purified N1ICD and MAML proteins at 2μM. Acrylamide gels were imaged using the LICOR Odyssey CLx scanner.

**Table 1. Key resources: All used materials including primer and antibodies.**

| Reagent or Resource | Source | Identifier |
|---|---|---|
| Antibodies | | |
| Goat Anti-Mouse IgM-UNLB | Southern Biotech | 1021–01 |
| Goat Anti-Mouse IgM-HRP | Southern Biotech | 1021–05 |
| Lyophilized Rabbit Polyclonal Antibody Ki67 Antigen | Novocastra | NCL-Ki67p |
| Cleaved Caspase-3 (Asp175) (5A1E) Rabbit mAb | Cell signaling | 9664P |
| Phospho-Histone H3 (Ser10) Antibody | Cell signaling | 9701S |
| Purified anti-Pax-5 Antibody | Biolegend | 649702 |
| Cy3-AffiniPure Donkey Anti-Rabbit IgG (H+L) | Jackson ImmunoResearch | 711–165–152 |
| Cy3 AffiniPure Donkey Anti-Rat IgG (H+L) | Jackson ImmunoResearch | 712–165–150 |
| Alexa Fluor 594 anti-mouse/human CD45R/B220 [RA3-6B2] | Biolegend | 103254 |
| Anti-Metallophilic Macrophages antibody [MOMA-1] | Abcam | ab51814 |
| Alexa Fluor 488-AffiniPure F(ab')2 Fragment Donkey Anti-Rabbit IgG (H+L) | Jackson ImmunoResearch | 711–546–152 |
| Alexa Fluor 647 anti-mouse IFN-γ [XMG1.2] | Biolegend | 505816 |
| Biotin anti-mouse IL-5 [TRFK4] | Biolegend | 504401 |
| Alexa Fluor 488 anti-mouse IL-17A [TC11-18H10.1] | Biolegend | 506909 |
| Alexa Fluor 647 anti-mouse CD3ε Antibody | Biolegend | 152319 |
| Anti-Cytokeratin 8+18 antibody (ab194130) | Abcam | ab194130 |
| Donkey anti-Guinea Pig IgG, Alexa -2788875 | Millipore Sigma | AP193SA6 |
| Mouse NKp46/NCR1 Antibody | R&D Systems | AF2225-SP |
| Alexa Fluor 647 AffiniPure F(ab')2 Fragment Donkey Anti-Goat IgG (H+L) | Jackson ImmunoResearch | 705-606-147 |
| anti-mouse/human CD45R/B220 [RA3-6B2] | Biolegend | 103206 |
| anti-mouse CD23 [B3B4] | Biolegend | 103222 |
| anti-mouse CD21/CD35 (CR2/CR1) [7E9] | Biolegend | 123416 |
| anti-mouse CD93 (AA4.1) | Biolegend | 136505 |
| anti-mouse IgM [RMM-1] 50 μg | Biolegend | 406505 |
| anti-mouse IgD [11-26c.2a] | Biolegend | 405723 |
| anti-mouse CD45.1 [A20] | Biolegend | 110706 |
| anti-mouse CD45.2 [104] | Biolegend | 109814 |
| Anti-Mouse CD3e Clone 145-2C11 | BD Biosciences | 553063 |
| anti-mouse Ly-6G [1A8] | Biolegend | 127615 |
| anti-mouse/human CD11b [M1/70] | Biolegend | 101216 |
| anti-mouse/human CD45R/B220 | Biolegend | 103227 |
| anti-mouse TCR β chain [H57-597] | Biolegend | 109208 |
| APC/Cy7 anti-mouse CD45R/B220 | Biolegend | 103224 |
| anti-mouse CD5 [53–7.3] | Biolegend | 100626 |
| Rat Anti-Mouse CD19 | Biolegend | bdb563557 |
| CD4 Monoclonal Antibody (RM4-5), PerCP-Cyanine5.5 | eBiosciences | 45-0042-82 |

(Continued)

**Table 1.** (Continued)

| Reagent or Resource | Source | Identifier |
| --- | --- | --- |
| CD8a Monoclonal Antibody (53–6.7), FITC | eBiosciences | 11-0081-82 |
| CD8a Monoclonal Antibody (53–6.7), PE-Cyanine7 | eBiosciences | 25-0081-82 |
| TCR beta Monoclonal Antibody (H57-597), APC-eFluor 780 | eBiosciences | 47-5961-82 |
| APC anti-mouse CD25 Antibody (clone PC61) | Biolegend | 102011 |
| CD4 Monoclonal Antibody (GK1.5), PE | eBiosciences | 12-0041-82 |
| PE/Cy7 anti-mouse CD117 (c-Kit) Antibody | Biolegend | 105814 |
| CD3e Monoclonal Antibody (145-2C11), FITC | eBiosciences | 11-0031-82 |
| CD19 Monoclonal Antibody (eBio1D3 (1D3)), PE | eBiosciences | 12-0193-82 |
| Notch2 (D76A6) XP | Cell signaling | 5732 |
| monoclonal anti-b-actin (AC-15) | Sigma-Aldrich | A5441 |
| ECL Rabbit IgG, HRP-linked whole AB (from Donkey) | GE Healthcare | NA934 |
| ECL Mouse IgG, HRP-linked whole AB (from Donkey) | GE Healthcare | NA931 |
| Bacterial and Virus Strains | | |
| BL21 (DE3) Tuner Competent cells | Novagen | 70623 |
| Biological Samples | | |
| Chemicals, Peptides, and Recombinant Proteins | | |
| XbaI | NEB | R0145 |
| BglII | NEB | R0144 |
| Mouse IgM-UNLB | Southern Biotech | 0101–01 |
| DyLight649(Cy5)-conjugated Streptavidin | Jackson Immunoresearch | 016-490-084 |
| Dextran sulfate sodium salt (DSS, M.W. approximately 36,000–50,000, colitis grade) | MP Biomedicals | 0216011050 |
| Red blood cell lysis buffer (10x) | BioLegend | 420301 |
| BSA | Fisher Scientific | Bp1600-100 |
| FBS | HyClone | AE24573269 |
| FBS | Sigma-Aldrich | F2442 |
| Nunc MaxiSorp flat bottom | Thermo Fisher Scientific | 44-2404-21 |
| TMB substrate solution | Thermo Fisher | N301 |
| RPMI-1640 | Sigma-Aldrich | R0883 |
| Gentamicin | Gibco B.R.L | 15750–060 |
| β-mercaptoethanol | Sigma-Aldrich | M7522 |
| U-bottom 96-well | Thermo Scientific | 174925 |
| Superfrost Plus microscope slides | Fisher Scientific | 12-550-15 |
| Xylene | Fisher Scientific | X3P |
| Alcian Blue 8GX | Sigma-Aldrich | A3157 |
| Nuclear Fast Red | Vector/Sigma-Aldrich | H-3403/N3020 |
| Trilogy | Cell Marque | 920P-10 |
| Normal donkey serum | Jackson ImmunoResearch | 017-000-121 |

(*Continued*)

**Table 1.** (Continued)

| Reagent or Resource | Source | Identifier |
|---|---|---|
| Prolong Gold Antifade Reagent | Invitrogen | P36930 |
| Complete Protease Inhibitor Cocktail | Roche | 11697498001 |
| Red Blood cell lysis buffer | BioLegend | 420301 |
| ACK lysis buffer | Lonza | 10-548E |
| Cell strainer (40 μm) | Falcon | 08-771-1 |
| Cell strainer (70 μm) | Falcon | 08-771-2 |
| Critical Commercial Assays | | |
| Supersignal West Femto Chemiluminescent substrate kit | Thermo Scientific/Fisher Scientific | P134095 |
| PureLink RNA Mini kit | Life Technologies (Invitrogen) | 12183025 |
| Superscript II reverse transcriptase | Invitrogen | 18064014 |
| iTaq Universal SYBR Green Supermix | Bio-Rad | 1725121 |
| Zombie VioletTM Fixable Viability kit | BioLegend | 423113 |
| Micro RNAeasy kit | Qiagen | 74106 |
| Zymo DNA Clean and Concentrator kit | Zymoresearch | D4029 |
| Deposited Data | | |
| SplitDam data | [45] | GEO: GSE70402 |
| ATAC-Seq data | This paper | GEO: GSE149992 |
| Experimental Models: Cell Lines | | |
| mK4 cells | [93] | N/A |
| Experimental Models: Organisms/Strains | | |
| N1RA | This paper | N/A |
| N2RA | This paper | N/A |
| N1 deletion | [94] | Notch1^tm1Con |
| N2 lacZ | [95] | N/A |
| C57BL/6 | Charles River | 027 |
| FVB/N | Harlan | N/A |
| B6D2F2 | Harlan | N/A |
| Oligonucleotides | | |
| N1-deletion fwd: ACGATATCGTGGTGCATACGCTCCTGTGCG | [94] | N/A |
| N1-deletion rev: GTCAGTTTCATAGCCTGAAGAACG | [94] | N/A |
| N1RA fwd: GCGGGATTCCTAGCCTGGTTACTC | This paper | N/A |
| N1RA rev: GTCCTTGTTGGCTCCGTTCTTCAG | This paper | N/A |
| N2-LacZ fwd: GTTGCAGTGCACGGCAGATACACTTGCTGA | [95] | N/A |
| N2-lacZ rev: GCCACTGGTGTGGGCCATAATTCAATTCGC | [95] | N/A |
| N2RA fwd: ACGGGCTCATCCCTGACATGAG | This paper | N/A |
| N2RA rev: TCCTCTCAGAAGGGTAGCAAGTC | This paper | N/A |
| Actb qPCR fwd: GGCTGTATTCCCCTCCATCG | Harvard PrimerBank | 6671509a1 |
| Actb qPCR rev: CCAGTTGGTAACAATGCCATGT | Harvard PrimerBank | 6671509a1 |

(*Continued*)

Table 1. (Continued)

| Reagent or Resource | Source | Identifier |
|---|---|---|
| Fos qPCR fwd: CGGGTTTCAACGCCGACTA | Harvard PrimerBank | 6753894a1 |
| Fos qPCR rev: TTGGCACTAGAGACGGACAGA | Harvard PrimerBank | 6753894a1 |
| Gapdh qPCR fwd: AGGTCGGTGTGAACGGATTTG | Harvard PrimerBank | 6679937a1 |
| Gapdh qPCR rev: TGTAGACCATGTAGTTGAGGTCA | Harvard PrimerBank | 6679937a1 |
| Myb qPCR fwd: AGACCCCGACACAGCATCTA | Harvard PrimerBank | 19526459a1 |
| Myb qPCR rev: CAGCAGCCCATCGTAGTCAT | Harvard PrimerBank | 19526459a1 |
| Notch1 qPCR fwd: GATGGCCTCAATGGGTACAAG | Harvard PrimerBank | 13177625a1 |
| Notch1 qPCR rev: TCGTTGTTGTTGATGTCACAGT | Harvard PrimerBank | 13177625a1 |
| Notch2 qPCR fwd: ATGTGGACGAGTGTCTGTTGC | Harvard PrimerBank | 33859592a1 |
| Notch2 qPCR rev: GGAAGCATAGGCACAGTCATC | Harvard PrimerBank | 33859592a1 |
| Pax5 qPCR fwd: CCATCAGGACAGGACATGGAG | Harvard PrimerBank | 6679213a1 |
| Pax5 qPCR rev: GGCAAGTTCCACTATCCTTTG | Harvard PrimerBank | 6679213a1 |
| IL1b-qPCR-fwd: CAACCAACAAGTGATATTCTCCATG | [96] | N/A |
| IL1b-qPCR-rev: GATCCACACTCTCCAGCTGCA | [96] | N/A |
| IL4-qPCR-fwd: AGATGGATGTGCCAAACGTCCTCA | [97] | N/A |
| IL4-qPCR-rev: AATATGGGAAGCACCTTGGAAGCC | [97] | N/A |
| Claudin4-qPCR-fwd: GTCCTGGGAATCTCCTTGGC | Harvard PrimerBank | 6753440a1 |
| Claudin4-qPCR-rev: TCTGTGCCGTGACGATGTTG | Harvard PrimerBank | 6753440a1 |
| ZO1-qPCR-fwd: GCACCATGCCTAAAGCTGTC | [98] | N/A |
| ZO1-qPCR-rev: ACTCAACACACCACCATTGC | [98] | N/A |
| TNFa-qPCR-fwd: CCCTCACACTCAGATCATCTTCT | Harvard PrimerBank | 7305585a1 |
| TNFa-qPCR-fwd: GCTACGACGTGGGCTACAG | Harvard PrimerBank | 7305585a1 |
| IL17A-qPCR-fwd: GACGCGCAAACATGAGTCC | This paper | N/A |
| IL17A-qPCR-rev: TTTGAGGGATGATCGCTGCT | This paper | N/A |
| N1RA sgRNA fwd: caccgCATTCGGGCATCCAGATCTG | This paper | N/A |
| N1RA sgRNA rev: aaacCAGATCTGGATGCCCGAATGc | This paper | N/A |
| N2RA sgRNA fwd: caccGGCATCCAGATCGGTTACA | This paper | N/A |
| N2RA sgRNA rev: aaacTGTAACCGATCTGGATGCC | This paper | N/A |
| N1RA donor oligo:<br>g*c*t*tgcatttagatcaccctgctgaaccatcctgcttccagaaccgcagctactgatctagatgccgaatgcatgatggcacaactccactgatcctggctgcggccctggccgtg*g*a*g | This paper | N/A |
| N2RA donor oligo:<br>a*g*c*caggcgggcagccaggatcagggggtagtaccatcgttcattctggcatccagatcgttaccgcgttgcggatcagaatctagaagagagaagcagagaagtgtcccttagataaggaaaaga*a*t*g | This paper | N/A |
| 2xCSL(EMSA): cgaaCGTGGGAAacctagctagaggcacCGTGGGAAaactagtgcgggcgtggct | This paper | N/A |
| 1xSPS(EMSA): gctaCGTGGGAAaggagcaaactgcgtTTCCCACGttccgtagtgcgggcgtggct | This paper | N/A |
| 5'IRDye-700_complementary_oligo(EMSA): agccacgcccgcact | This paper | N/A |
| 5'IRDye-800_complementary_oligo(EMSA): agccacgcccgcact | This paper | N/A |
| Recombinant DNA | | |
| Plasmid: N1AE | [30] | N/A |

(Continued)

**Table 1.** (Continued)

| Reagent or Resource | Source | Identifier |
|---|---|---|
| Plasmid: N1RAΔE | This paper | N/A |
| pSpCas9(BB) | [99] | 48139 |
| pTXB1-Tn5 | [92] | N/A |
| Software and Algorithms | | |
| Image lab | Bio-Rad Laboratories | https://www.bio-rad.com/en-us/product/image-lab-software?ID=KRE6P5E8Z |
| Prism 8 | Graph Pad | www.graphpad.com |
| CCTop tool | [82] | https://crispr.cos.uni-heidelberg.de |
| NIS-Elements Advanced Research software | Nikon | N/A |
| FlowJo software v9.7 | Becton Dickinson | https://www.flowjo.com |
| Other | | |
| LPS from *Salmonella enterica* | Sigma-Aldrich | L6143 |
| HDM extract (*Dermatophagoisdes farina*) | Greer Laboratories Inc./Fisher Scientific | NC0277827 |

AB, antibody; ACK, ammonium-chloride-potassium; BSA, bovine serum albumin; CSL, CBF1/Suppressor of Hairless/LAG-1; DSS, dextran sodium sulfate; EMSA, electrophoretic mobility shift assay; FBS, fetal bovine serum; HDM, house dust mite; HRP, horseradish peroxidase; IgG, immunoglobulin G; IgM, immunoglobulin M; LPS, lipopolysaccharide; mAb, monoclonal antibody; N1, Notch1; N1ΔE, constitutive active Notch1 lacking extracellular domain; N2, Notch2; qPCR, quantitative PCR; RA, Arg-Ala substitution; RPMI, Roswell Park Memorial Institute; sgRNA, short guide RNA; SPS, sequence-paired site; TMB, 3,3′,5,5′-Tetramethylbenzidine; UNLB, unlabeled.

## EMSA quantitation

To quantitatively analyze the EMSA experiments, we first extracted the grayscale intensity of each band in each lane of the EMSA gels (S1A and S1B Fig). We then fitted the data to a 2-site binding model that takes into account cooperative binding to the second site. The model calculates the binding probability assuming equilibrium binding kinetics (Michaelis-Menten) to 2 sites. Cooperativity is taken into account by assuming that the binding dissociation constant of a complex to the second site is divided by a cooperativity factor $C$, namely that $K_{d,2nd} = \frac{1}{C}K_{d,1st}$. The probabilities to find the probe bound by 0, 1, or 2 complexes are given by

$$P_0 = \frac{1}{1 + 2\alpha + C\alpha^2}, P_1 = \frac{2\alpha}{1 + 2\alpha + C\alpha^2}, P_2 = \frac{C\alpha^2}{1 + 2\alpha + C\alpha^2}$$

where $\alpha = \frac{[NTC]}{K_d}$ is the statistical weight associated with binding of the NTC complex to a CSL site, and $K_d$ is the dissociation constant to a single site. If the cooperativity factor, $C$, is equal to 1, then the binding to the 2 sites is noncooperative. $C>1$ corresponds to positive cooperativity (second binding is enhanced). $C<1$ corresponds to negative cooperativity (second binding is suppressed).

Because we observe that even at high concentrations of RBPj, the 1-site state is never depleted (e.g., see RBPj on CSL), we assumed that there is only some fraction of the probes, $f<1$, that binds 2 complexes and another fraction, $(1-f)$, that can only bind 1 complex. In this case, the probability to find the probe is modified to

$$P_0 = f\frac{1}{1 + 2\alpha + C\alpha^2} + (1-f)\frac{1}{1 + 2\alpha}, P_1 = f\frac{2\alpha}{1 + 2\alpha + C\alpha^2} + (1-f)\frac{2\alpha}{1 + 2\alpha}, P_2$$
$$= f\frac{C\alpha^2}{1 + 2\alpha + C\alpha^2}$$

We then fit the experimental data (normalized band intensities) to these expressions. The fitting parameters are $K_d$, $C$, and $f$. The parameters are extracted for each experiment separately.

To get an estimation of the confidence interval on the fitting parameters, we use a bootstrap method to randomly generate 5,000 data sets with the same statistical properties as the experimental data sets (the same mean and standard deviation of band intensities). We then perform the same fitting procedure on all bootstrapped data to obtain the 95% confidence interval on the fitting parameters.

## Flow cytometry

For materials used, see Key resource table (Table 1). MZB and FoB cells were sorted by FACS as described in [87]. Single-cell splenocytes were prepared by placing the spleen in a petri dish and mincing it with a razor blade. The disrupted spleen tissue was then transferred into a 50-ml falcon tube and pipetted up and down in 5 ml of ice-cold 1% BSA (Fisher Scientific) in PBS and briefly vortexed to ensure thorough disruption. Single-cell suspensions were obtained by passing the disrupted spleen tissue through a 70-μm strainer (Falcon). The strainer was then rinsed with 5 ml ice-cold 1% BSA/PBS and the single-cell preparation pelleted by centrifuging at 2,000 rpm, for 5 minutes followed by red blood cell lysis (BioLegend), after which the cells were pelleted at 2,000 rpm, for 5 minutes, and resuspended in ice-cold 3% BSA in PBS before cell counting. Single-color antibody stained as well as unstained controls were prepared with 3 million splenocytes each, 1.5 μl (0.75 μg) of each antibody was added into the corresponding single-color control tube. For MZB detection, approximately 10 million cells

(in 1 ml 3% BSA) were distributed into Eppendorf tubes, and 7.5 μl of the antibody mix (2.5ul each of FITC anti-B220, PE/Cy7 anti-CD23 and PerCP/Cy5.5 anti-CD21) were added into each tube. They were then incubated at 4˚C for 60 minutes before sorting for B220$^+$CD21$^{hi}$CD23$^{lo}$ MZB cells and B220$^+$CD21$^{int}$CD23$^{hi}$ FoB cells. To isolate MZB (CD93$^-$B220$^+$CD19$^+$Ig-M$^+$IgD$^+$CD23$^{lo}$), FoB (CD93$^-$B220$^+$CD19$^+$IgM$^+$CD21$^+$), MZP (CD93$^-$B220$^+$CD19$^+$CD21$^{hi}$IgM$^{hi}$IgD$^{hi}$CD23$^{hi}$), and T2 B-cells (CD93$^+$B220$^+$IgM$^+$CD23$^+$), single-cell splenocytes were incubated with the APC/Cy7 anti-B220, PE/Cy7 anti-CD93, Hoechst blue anti-CD19, FITC anti-IgM, AmCyan anti-IgD, Percp/Cy5.5 anti-CD21 and Pacific blue CD23 and FACS sorted. To analyze peritoneal B1 B-cell populations, peritoneal cells were collected as described in [88] and incubated with APC/Cy7 anti-B220, PE anti-TCRb, FITC anti-CD11b, PE/Cy7 anti-CD23, and APC anti-CD5. They were then gated into B2 B-cells (B220$^+$CD23$^+$) and B1 B-cells (B220$^+$CD23$^-$). B1 B-cells were then gated into B1a (CD11b$^+$CD5$^+$) and B1b (CD11b$^+$CD5$^-$) B-cells.

For T-cell analysis, thymus and spleen were harvested and disrupted by gentle grinding of tissue with a pestle (CellTreat) on a 40-μm cell strainer (Falcon). Red blood cells in spleen samples were lysed using ACK lysis buffer (Lonza) for 5 minutes at 4˚C. Cells were counted and stained at 20×106 cells/ml. Cells were resuspended in PBS with Zombie Violet viability dye (Biolegend) and incubated at 4˚C for 20 minutes. Cells were washed and then resuspended in an antibody cocktail in PBS supplemented with 2% FBS (Hyclone) in the presence of 5% 2.4G2 Fc blocking antibody (in house) and incubated for 30 minutes at 4˚C. Cells were washed, resuspended in PBS with 2% FBS, and analyzed using an LSRII (BD). Data were analyzed using FlowJo software v9.7, and average population percentages and absolute numbers were graphed using GraphPad Prism (GraphPad Software, Inc.).

## MZB and FoB culture and ELISA

For materials used, see Key resource table (Table 1). MZB and FoB cells were cultured in RPMI-1640 (Sigma-Aldrich) containing 10% FBS (Sigma-Aldrich), 10 μg/ml gentamicin (Gibco B.R.L) and 50 μM β-mercaptoethanol (Sigma-Aldrich). A total of 10$^5$ MZB and FoB cells were seeded into U-bottom 96-well plates (Thermo Scientific) in 200μL of media. Differentiation was stimulated by adding lipopolysaccharides from *Salmonella enterica* serotype typhimurium -LPS (Sigma-Aldrich) into each well at a final concentration of 2 μg/ml. The LPS-stimulated cells were cultured at 37˚C, 5% $CO_2$, for 5 days. To measure IgM levels by ELISA, MaxiSorp flat-bottom 96-well plates (Nunc) were coated with 200 μL of 5 μg/ml goat anti-mouse IgM (Southern Biotech) in 0.05% PBST (PBS-tween20) at 4˚C, overnight. The plates were then washed thrice with 0.05% PBST and blocked with 3% BSA (Fisher Scientific) in PBS for 30 minutes at room temperature. Purified mouse IgM (Southern Biotech) was used to prepare a control standard. A total of 150 μl of supernatant collected from the stimulated B-cells at the end of 5-day incubation was diluted in 0.05% PBST (PBS containing 0.05% Triton-100), and along with the standard, was loaded on to the MaxiSorp flat-bottom (Thermo Fisher Scientific) ELISA plates for 1-hour incubation at room temperature followed by 3 washes with PBST. HRP goat anti-mouse IgM (Southern Biotech), diluted at 1:5,000 in PBST, was added and incubated for 1 hour at room temperature followed by 3 washes with PBST. TMB substrate solution (Thermo Fisher Scientific) was then added and developed for 10 minutes. Development was stopped with 0.16 M sulfuric acid before colorimetric absorption analysis at 450 nm.

## Histology

For materials used, see Key resource table (Table 1). Mouse tissues were fixed overnight in 4% PFA and embedded in paraffin. Adult intestinal tissue was divided into sections corresponding

to the jejunum, duodenum, ileum, and colon. Each section was then folded into a swiss-roll [89] and then fixed, mounted and sectioned for histological analysis. For P0 animals, whole intestinal tissue was collected and then fixed and mounted whole before sectioning for histological or immunofluorescence analysis. Sections 5-μm thick were deposited on Superfrost plus microscope slides (Fisher Scientific), deparaffinized with xylene (Fisher Scientific), and rehydrated. Sections were stained with hematoxylin for 2.5 minutes and Eosin for 1 minute before dehydration, clearing, and mounting. For Alcian blue staining, sections were rinsed in 3% acetic acid (Fisher Scientific) for 3 minutes and incubated at room temperature for 30 minutes in 1% Alcian blue 8GX (Sigma-Aldrich) in 3% acetic acid solution. The sections were then rinsed for 3 minutes in 3% acetic acid followed by 10 minutes in running water. After counterstaining with Nuclear Fast Red (Vector) for 5 minutes, slides were rinsed for 10 minutes in running water prior to dehydration, clearing, and mounting. For each sample, images of at least 10 random fields of view were acquired on a Nikon 90i Upright widefield microscope. Where indicated, quantitative analysis of images was done using Nikon's NIS-Elements Advanced Research software (NIS-AR). In order to automatically identify and quantify Alcian blue positive goblet cells, an inhouse NIS-AR plugin was applied to identify and mask the discrete, large, dark blue Alcian blue spots scattered on the pink-white Nuclear Fast Red counterstain (Fig 3). We then manually traced the villi in the region of interest (ROI) to exclude the Alcian blue stained mucus. The plugin then counted the Alcian blue positive foci within the ROI. Goblet cell numbers were determined for all acquired fields of view for each animal analyzed. The average number of goblet cells per field of view was obtained for each analyzed animal. Where multiple animals of the same genotype were analyzed, the average goblet cell number per field of view was obtained by averaging the goblet cell numbers in all fields of view from all the animals of that genotype.

## Fluorescent immunohistochemistry

For materials used, see Key resource table (Table 1). Paraffin embedded sections were deparaffinized with xylene, rehydrated, and subjected to heat-induced epitope retrieval (HIER) by boiling in Trilogy (Cell Marque) for 30 minutes. After cooling, the sections were rinsed with distilled water for 5 minutes followed by PBS with 0.3% Triton-X100 (Fisher Scientific) for 5 minutes. The sections were then incubated at room temperature for 2 hours in blocking solution (10% normal donkey serum [Jackson ImmunoResearch] in PBS-Triton), followed by overnight incubation with respective primary antibodies at 4˚C. The sections were washed 3 times for 10 minutes each (3×10 minutes) with PBS-Triton and incubated for 2 hours at room temperature with Fluorophore-labeled secondary antibodies in blocking solution. Unbound secondary antibody was washed 3×10 minutes in PBS-Triton and where indicated, counterstained with DAPI before mounting with Prolong Gold Antifade (Invitrogen). Confocal imaging was performed on a Nikon Ti-E Inverted Microscope.

## Cytoplasmic/Nuclear separation and Western Blot

For materials used, see Key resource table (Table 1). N2ICD stability was determined in FACS-isolated MZB and FoB cells. Sorted cells were pelleted for 5 minutes with 2,000 rpm at 4˚C and resuspended in 500-μl hypotonic buffer (HB; 20 mM Tris-HCl [pH 7.5], 10 mM NaCl, 3 mM $MgCl_2$) with added protease inhibitor (Roche). After 30 minutes' incubation on ice, 25 μl of 10 NP-40 was added, and samples were vortexed for 10 seconds. Nuclei were pelleted at 3,000 rpm for 10 minutes at 4˚C, and the cytoplasmic fraction transferred into a fresh tube. The nuclei pellets were washed twice in 500 μl HB and then lysed in 20 μl 2X Laemmli sample buffer (120 mM Tris-HCl [pH 6.8], 20% glycerol, 4% SDS). The cytoplasmic fractions

were concentrated using AmiconUltra– 0.5 mL– 30K centrifugal filters (Merck Millipore) and mixed 1:1 with 2X Laemmli sample buffer. The extracts were separated on 6% polyacrylamide (Bio-Rad) gel and transferred onto nitrocellulose membranes in Tris/glycine transfer buffer. Membranes were blocked with blocking solution (5% milk in 0.1% PBS-Tween20 [Fisher Scientific]) for 1 hour, room temperature, and incubated overnight at 4˚C with primary antibody in blocking solution. Membranes were then washed 3 times for 5 minutes with 0.1% PBS-Tween20 and incubated for 1 hour at room temperature with anti-rabbit HRP secondary antibody (GE healthcare) in blocking solution. Membranes were developed with a Supersignal West femto chemiluminescent substrate kit (Fisher scientific) and developed using Chemidoc (Bio-Rad) detection system. Signal intensities were quantified using Image Lab (Bio-Rad) software.

## qPCR

For materials used, see Key resource table (Table 1). RNA from tissue and cells was extracted using the PureLink RNA Mini kit (Invitrogen) and cDNA was synthesized with SuperScript II reverse transcriptase (Invitrogen) following the manufacturers instruction. Quantitative PCRs were performed using iTaq Universal SYBR Green Supermix (Bio-Rad) on the StepOnePlus RT PCR system. Data were analyzed using the Delta-Delta-CT methods. A full list of oligos is provided in In Key Resources Table.

## ATAC-Seq

For materials used, see Key resource table (Table 1). For sample library preparation, we followed the Omni-ATAC method outlined by [90, 91] and purified Tn5 was generated as described [92]. Briefly, 50,000 nuclei from FACS-sorted MZB cells were processed for Tn5 transposase-mediated tagmentation and adaptor incorporation at sites of accessible chromatin. FACS-isolated MZB cells were pelleted and washed with ice-cold PBS. The pellet was resuspended in ATAC-Resuspension Buffer (10 mM Tris-HCl [pH 7.4], 10 mM NaCl, 3 mM MgCl$_2$ 0.1% NP40, 0.1% Tween-20, 0.01% Digitonin) and incubated on ice for 3 minutes. The lysed cells were washed in ATAC-Wash Buffer (10 mM Tris-HCl [pH 7.4], 10 mM NaCl, 3 mM MgCl$_2$, 0.1% Tween-20), inverted 3 times, and the nuclei pelleted. The nuclei were resuspended in ATAC transposition mix (10 mM Tris-HCl [pH 7.6], 10 mM MgCl$_2$, 20% Formamide, 100 nM Tn5 transposase) and incubated at 37˚C for 30 minutes in a thermomixer at 1,000 RPM. Following tagmentation, the DNA fragments were purified using a Zymo DNA Clean and Concentrator Kit, and library amplification was performed using customized Nextera PCR primer Ad1 in combination with any of Ad2.1 through Ad2.12 barcoded primers as described [91]. The quality of the purified DNA library was performed utilizing an Agilent Bioanalyzer 2100 using High Sensitivity DNA Chips (Agilent Technologies Inc., Santa Clara, CA). The samples were pooled at a concentration of 5 nM and run on an Illumina HI-SEQ 2500 sequencer (Illumina, Inc. San Diego, CA) to obtain paired-end reads of 75 bases (PE75). ATAC sequencing was carried out on 2 conditions MZB WT and RA. For each condition, 4 biological replicates were sequenced, in single-end fashion. Read depth for replicates varied from approximately 50 to 98 million reads thereby giving us average read depth of approximately 67 million reads. Quality check, mapping, and peak calling was performed using CSBB-v3.0 [Process-ChIP-ATAC_SingleEnd]. CSBB uses fastqc, bowtie2 and macs2 [parameters:—nomodel—shift 37—extsize 73], respectively. Duplicate mapped reads were removed before peak calling. Mapped reads in bam format was converted to bigwigs using deeptools (deeptools.ie-freiburg.mpg.de) for visualization purposes. For assessing open nucleosome region differences from MZB WT to RA MZB, we performed differential peak analysis. Within each sample type, only peaks with 75% replication and at least 50% overlap among biological

replicates were used for differential peak analysis. Further, peak sets passing above defined criteria were merged (at least 1-bp overlap) using bedtools merge and then number of reads mapping under each peak for each replicate was inferred using FeatureCounts program. Finally, differential peak analysis was performed using EdgeR. No statistically significant changes in peaks (ATAC regions) between WT MZB and RA MZB were identified.

## Supporting information

**S1 Fig. Loss of Notch1 dimerization suppresses target gene expression in MK4 cells (S1 Fig, see S1 Data for raw data).** (A-B) EMSA data for purified proteins binding to probes containing 2 CSL (A) or SPS (B) sites. The number of balls marks the occupancy of sites; arrows below indicate orientation of binding sites in probes. (C-D) The average sites filled were plotted against RBPj concentrations to calculate the cooperativity factor "C" on CSL (C) and SPS (D) probes. (E) Relative expression of Notch target genes in mK4 cells overexpressing either N1ΔE or N1$^{RA}$ΔE as determined by RNA Seq (all replicates are shown, values ranked per row. Green arrows indicate targets elevated more in N1$^{RA}$ΔE, red arrows indicate targets repressed in in N1$^{RA}$ΔE relative to control. (F) Luciferase activation assays were used to analyze the homo- and heterodimerization properties of Notch1 and Notch2 on a dimer-dependent reporter gene (Hes5-Luciferase) and a dimer-agnostic reporters (Hey2-Luciferase). As previously shown [38], the dimer interface involves 3 amino acids, the Arg (R) mutated in this study, as well as the positively charged Lys (K1935, Notch1) and a negatively charged Glu (E1939, Notch1, or Asp [D1899] at the equivalent position in Notch2) that form salt bridges between NICD molecules. Dimer formation is critical for Notch-dependent activation of the Hes5 reporter, and mutating any of these amino acids abrogates activation. Notably, by co-expressing 2 Notch protein with complementary mutations, one in which K1935 was changed to E (N1KE), and a second in which D1899 was changed to K (N2DK) leads to strong enhancement in Luciferase expression, explained by achieving a more favorable conformation of the complementing mutant dimer. Note synergistic complementation between N1KE and N2DK but not between N1KE and N2KD, providing strong evidence of a cooperative heterodimer. Significance tested by a Student $t$-test. CSL, CBF1/Suppressor of Hairless/LAG-1; EMSA, in electrophoretic mobility shift assay; NICD, notch intracellular domain; N1$^{RA}$ΔE, extracellular domain deleted Notch1 Arg$^{1974}$Ala mutant; RPBj, Recombinant binding protein for immunoglobulin Kappa j region; SPS, sequence-paired site.
(PDF)

**S2 Fig. Normal development of T-cell compartments in _N1$^{RA/RA}$_ mice (S1 Fig, see S1 Data for raw data).** Thymi and spleens were isolated from _wt_, _N1$^{+/RA}$_, or _N1$^{RA/RA}$_ mice, and the T-cell compartment was analyzed. (A) The absolute number of thymic single- and double-positive T cells (left) and developing T cells (right) was assessed by flow cytometry. (A') The average number of cells in the thymus (left) and average thymic weight (right) are shown. (B) The percentage (left) and absolute number (right) of T-cell subsets in the spleen were assessed by flow cytometry. (B') The average number of splenocytes (left) and average spleen weight (right) are shown. (_n_ = 3–6 mice per genotype; error bars = +/- SEM). _N1$^{+/RA}$_, Notch1 Arg$^{1974}$Ala heterozygote; _N1$^{RA/RA}$_, Notch1 Arg$^{1974}$Ala homozygote; _wt_, wild-type.
(PDF)

**S3 Fig. Proliferation potential in wt versus mutant intestines with or without fur mite exposure (S4 Fig).** Sections of P0 and P1 intestines from animals born to fur mite–infested dams (A-H') and born after fur mite eradication (I-J') were stained for Ki67 (A-D', I, and I') and phospho-H3 (E-H', J, and J') to assess for proliferation in the crypts. _N1$^{+/-}$;N2$^{+/-}$_ are

indistinguishable from *wt* (A-B', E-F'), whereas proliferation is significantly reduced in $N1^{+/-}$; $N2^{RA/RA}$ and $N1^{RA}/-;N2^{RA/-}$ crypts (C-D', G-H'). Proliferation in $N1^{RA/-};N2^{RA/-}$ is still decreased compared with heterozygous littermates after fur mite eradication (I-J'). $N1^{+/-};N2^{+/-}$, *Notch1/Notch2* RA hemizygous; $N2^{RA/RA}$, *Notch2* RA homozygous; P0-1, postnatal day 0–1; RA, Arg ($N1^{R1974}/N2^{R1934}$) to Ala substitution; wt, wild-type.
(PDF)

**S4 Fig. Like wild type MZB cells, $N2^{RA/RA}$ MZB cells have a robust proliferative response upon LPS stimulation and N2RA/RA spleen display germinative centers (S5 Fig).** Isolated MZB from $N2^{RA/RA}$ and *wt* littermates were cultured and stimulated with LPS. Both genotypes proliferate after stimulation (A). Sections of $N2^{RA/RA}$ and *wt* spleens were stained with Ki67 and phosphor-H3 to detect proliferation and Caspase3 for apoptosis (B). Proliferation in germinative centers were detected in dimer-deficient mice but not in *wt*. Apoptosis was not altered. LPS, lipopolysaccharide; MZB, marginal zone B-cell; *Notch2* RA homozygous; RA, Arg ($N1^{R1974}/N2^{R1934}$) to Ala substitution; wt, wild-type.
(PDF)

**S5 Fig. Spleen size as a function of age.** In the absence of pathogens; $N1^{+/RA};N2^{RA/RA}$ enlarged spleens and lymph nodes of mite-infested mice have a high proliferative and mitotic index (S6 Fig, see S1 Data for raw data). (A). In the absence of fur mites, an increase in spleen size with aging was observed in $N2^{RA/RA}$ mice (RA), but not in mice with other genotypes (+) housed in the same colony (B, D). The enlarged spleens and lymph nodes from aged $N1^{+/RA};N2^{RA/RA}$ mice showed increased staining for Ki67 and phosphor-H3 indicating proliferation Apoptosis was slightly increased in enlarged spleens as shown by Caspase3 stain (C). (E) Low magnification of spleens from aged $N2^{RA/RA}$ and $N1^{+/RA};N2^{RA/RA}$ mice infested with fur mites show expansion of white pulp. $N1^{+/RA};N2^{RA/RA}$, *Notch1* RA heterozygote, *Notch2* RA homozygous; RA, Arg ($N1^{R1974}/N2^{R1934}$) to Ala substitution.
(PDF)

**S6 Fig. Loss of NICD dimerization does not stabilize the protein (S6 Fig; see S1 Data for raw data).** Western blot analysis of nuclear (N) and cytoplasmic (C) preparations of sorted MZB shows no difference in N2 stability between *wt* and $N2^{RA/RA}$ mice (A); immunoblotting of α-tubulin and histone-h3 confirms separation of cytoplasmic and nuclear fractions, respectively. Quantification of the nuclear N2ICD relative to cytoplasmic total N2 reveals no difference between wt and dimer-deficient N2ICD stability (B). MZB, marginal zone B-cell; NICD, Notch intracellular domain; wt, wild-type.
(PDF)

**S1 Table. Chi-squared analysis of pups (P0 and P1) born in of $N1^{RA/RA};N2^{RA/RA}$ x $N1^{+/-}$; $N2^{+/-}$ cross in mite-free housing.** N1, Notch1; N2, Notch2; RA, Arg ($N1^{R1974}/N2^{R1934}$) to Ala substitution; P, postnatal day.
(PDF)

**S2 Table. Chi-squared analysis of male and female pups born in C57BL/6J $N1^{+/RA}$ x $N1^{+/RA}$ cross during fur mite infestation and following generations.** $N1^{+/RA}$, Notch1 Arg$^{1974}$Ala heterozygote.
(PDF)

**S1 Data. Original images and data used to generate figures and tables presented in the manuscript, organized in tabs corresponding to figure panels.**
(XLSX)

## Acknowledgments

The authors wish to thank Dr. Matt Kofron with help with imaging, Dr. Yueh-Chiang Hu and the genome editing core for generating the NRA alleles, and members of the Kopan lab for comments and encouragement during the study. Dr. Stephan Waggoner, Kelli VanDussen and members of the DB and Notch community for valuable advice. Many thanks to Dr Sai Tummala and the veterinarians at CCHMC.

## Author Contributions

**Conceptualization:** Eric W. Brunskill, Raphael Kopan.

**Data curation:** Kristina Preusse, Warren S. Pear, Rhett A. Kovall, David Sprinzak, Brian Gebelein, Eric W. Brunskill, Raphael Kopan.

**Formal analysis:** Francis M. Kobia, Kristina Preusse, Praneet Chaturvedi, David Sprinzak, Eric W. Brunskill, Raphael Kopan.

**Funding acquisition:** Raphael Kopan.

**Investigation:** Francis M. Kobia, Kristina Preusse, Quanhui Dai, Nicholas Weaver, Matthew R. Hass, Sarah J. Stein, Warren S. Pear, Zhenyu Yuan, Yi Kuang, Natanel Eafergen, Eric W. Brunskill.

**Supervision:** Kristina Preusse, Eric W. Brunskill, Raphael Kopan.

**Validation:** Francis M. Kobia.

**Visualization:** Raphael Kopan.

**Writing – original draft:** Francis M. Kobia.

**Writing – review & editing:** Kristina Preusse, Quanhui Dai, Matthew R. Hass, Rhett A. Kovall, Yi Kuang, David Sprinzak, Brian Gebelein, Eric W. Brunskill, Raphael Kopan.

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
