## [Editor Report · Decision Letter 0]

9 Mar 2020

Dear Raphael, 

Thank you for submitting your manuscript entitled "Exposure to Mites Sensitizes Intestinal Stem Cell Maintenance, Splenic Marginal Zone B Cell Homeostasis, And Heart Development to Notch Dosage and Cooperativity." for consideration as a Research Article by PLOS Biology.

Your manuscript has now been evaluated by the PLOS Biology editorial staff as well as by an academic editor with relevant expertise and I am writing to let you know that we would like to send your submission out for external peer review.

Please re-submit your manuscript within two working days, i.e. by Mar 11 2020 11:59PM.

Kind regards,

Ines

--

Ines Alvarez-Garcia, PhD

Senior Editor

PLOS Biology

Carlyle House, Carlyle Road

Cambridge, CB4 3DN

+44 1223–442810

---

## [Decision Letter · Decision Letter 1]

22 Apr 2020

Dear Rafi,

Thank you very much for submitting your manuscript "Exposure to Mites Sensitizes Intestinal Stem Cell Maintenance, Splenic Marginal Zone B Cell Homeostasis, And Heart Development to Notch Dosage and Cooperativity." for consideration as a Research Article at PLOS Biology. Thank you also for your patience as we completed our editorial process, and please accept my apologies for the delay in providing you with our decision. Your manuscript has been evaluated by the PLOS Biology editors, an Academic Editor with relevant expertise, and by two independent reviewers.

As you will see, both reviewers are positive and find the conclusions of your manuscript very interesting and significant for the field. Nevertheless, they have also raised a few issues that need to be addressed. After discussing the reviews with the Academic Editor and considering the current circumstances, we will not make essential the experiments suggested, such as the CHIP-seq analysis of Notch2+/+ and Notch2RA/RA, but we will welcome any data you might have in hand. Please address/discuss all the issues and highlight any unanswered questions in the discussion. 

In light of the reviews (attached below), we are pleased to offer you the opportunity to address the remaining points from the reviewers in a revised version that we anticipate should not take you very long. We will then assess your revised manuscript and your response to the reviewers' comments and we may consult the reviewers again.

We expect to receive your revised manuscript within 2 months.

**IMPORTANT - SUBMITTING YOUR REVISION**

*Resubmission Checklist*

*Published Peer Review*

*PLOS Data Policy*

*Blot and Gel Data Policy*

Best wishes,

Ines

--

Ines Alvarez-Garcia, PhD

Senior Editor

PLOS Biology

Carlyle House, Carlyle Road

Cambridge, CB4 3DN

+44 1223–442810

Reviewers’ comments

Rev. 1:

This manuscript is a deep, thorough, and fascinating exploration of the functional consequences of introducing single point substitutions into murine Notch1 and Notch2 that interfere with cooperative dimerization of Notch transcription complexes at head-to-head paired site elements (SPS sites). In isolation, neither dimer-deficient variant results in a developmental phenotype, nor does combining the two alleles in a double homozygote. Under environmental stress, however, N1 RA/RA, N2 RA/RA mice exhibit colonic barrier defects in a DSS colitis model. Analysis of N1 RA/-, N2 +/RA mice shows partial penetrance of a ventricular septal defect that is more pronounced in N1 RA/-, N2 RA/- mice, and fur-mite challenged N2 RA/RA mice show marginal zone B (MZB) cell expansion that can resemble MZB lymphoma in aged mice. Mechanistic studies to elucidate the origin of the paradoxical MZB gain-of-function phenotype in the N2 RA/RA mice suggests that selective loading of this N2 allele on monomer-responsive genes accounts for this effect. This manuscript certainly warrants publication in PLOS Biology after a few minor concerns are addressed.

Comments:

1. The mite infestation appears to have been an unplanned stressor of the genetically engineered mice. Did the authors evaluate whether the mites preferentially colonized the mutant mice compared with the wild-type mice?

2. In figure 5E, the authors make the point that HDM-induced dermatitis elevated MZB numbers only in the N2 RA/RA mice but not in controls (such as LPS-treated mice). The data do, however, show the same trend in the permethrin and LPS-treated mice - it is just that the data are somewhat noisy and the trends in the other conditions don't reach statistical significance. Could it be that the study was underpowered? The authors might make note of the trend, without undermining the overall message of the figure panel or the study.

3. For those readers who are not Notch cognoscenti, the authors should expand the text to clarify for the more general reader the split-dam methodology use to compare loading of Notch-RBPJ monomeric complexes and dimeric complexes (that have NICD-D/NICD-AM within the same complex in order to build a functional DAM molecule). They should also improve the quality of the figure legend (D/AM "haves" I think means D/AM "halves" - and the text could also be made clearer for a naïve reader).

Rev. 2:

Kobia et al. described the in vivo roles of cooperative DNA binding of intracellular domains of Notch receptors in heart, intestinal epithelium, and spleen. By meticulous phenotypical analysis of sophisticated knock-in mouse models of Notch1 and Notch2, authors found that dimerization-dependent Notch signaling is essential for heart development, injury response of colonic stem cells, and homeostasis of marginal zone B-cell. The most intriguing finding is Notch cooperativity can be induced at a certain specific context such as mite infestation, which is not preferred in a normal physiological status.

Dimer-deficient mice generated in this study are wonderful model system to investigate the differential roles between the conventional NTCs and the cooperative NTCs on SPS in vivo. The most of analysis were focused on phenotypical differences in various combination of dimer-compatible and dimer-defective alleles with different Notch dosages. Those are still very invaluable and informative resources to comprehend the fine-tuning mechanisms of diverse outcomes of Notch pathways. However, the underlying molecular mechanisms (provided mostly by Figure 7, only differential gene expression by RNA-seq analysis were examined) are not quite sufficient to explain the complex phenotypes of N1RA and N2RA alleles. Dissection of dimer-dependent and dimer-independent gene regulation using specific cell types affected in animal models (e.g. marginal zone B cells or intestinal stem cells) will greatly improve the manuscript. Detail comments are described below.

Major comments

- Perhaps, one of the most intriguing question would be where dimer-compatible NTC and dimer-defective NTC regulate gene expression. Do they regulate differential enhancer element in normal physiology and inflammatory environment (mite infestation or 1% DDS treated)? ATAC-seq analysis can be one of evidences, but it showed minimal changes in chromatin status between WT and N2RA/RA mice. SplitDAM experiments in Figure 7F is quite interesting but were performed in non-physiological context and only showed Myb locus. CHIP-seq analysis of Notch2+/+ and Notch2RA/RA can be a feasible approach to address this question.

- Where are inflammation-sensitive SPS sites contributing altered gene expression in dimer-defective animal models? How can those SPSs upregulate or downregulate Notch target genes? Providing few examples (eg. Myb, FoxM1, E2F1) would be very helpful to understand the potential molecular mechanism of differential Notch pathway outcomes.

Minor comments

- in page 8, "1/8 surviving P0 N1+/-; N2+/- pups" -> "1/8 surviving P0 N1RA/-; N2RA/- pups

- The labels in Figure 5B are improperly presented (overlapped).

---

## [Editor Report · Decision Letter 2]

3 Jun 2020

Dear Dr Kopan,

Thank you very much for submitting a revised version of your manuscript "Exposure to Mites Sensitizes Intestinal Stem Cell Maintenance, Splenic Marginal Zone B Cell Homeostasis, And Heart Development to Notch Dosage and Cooperativity." for consideration as a Research Article at PLOS Biology. This revised version of your manuscript has been evaluated by the PLOS Biology editors and by the Academic Editor.

We will probably accept the manuscript if you are willing to address the remaining issues highlighted by the academic editor regarding Figures 1 and 7 - please also find attached an annotated word file with suggestions. In addition, we feel the manuscript would benefit from a significant proofreading to fix potential errors.

**IMPORTANT - SUBMITTING YOUR REVISION**

*Resubmission Checklist*

*Published Peer Review*

*PLOS Data Policy*

Many thanks for including a data file containing all raw data summarised in the figures. Nevertheless, we are missing data from the following figures:

Fig. 2H; Fig. 4J and Fig. 7D (if you include it)

Please also ensure that both your main and supplementary figure legends in your manuscript include information on WHERE YOUR DATA CAN BE FOUND. Please ensure that your Data Statement in the submission system accurately describes where the underlying data can be found.

*Blot and Gel Data Policy*

Sincerely,

Ines

--

Ines Alvarez-Garcia, PhD

Senior Editor

PLOS Biology

Carlyle House, Carlyle Road

Cambridge, CB4 3DN

+44 1223–442810

Academic Editor's comments

In Figure 1:

The N2RA has not been fully characterized and there is lack of evidence that N2RA would show similar dimerization deficiency like N1RA. It is recommended to point this out either here or in Discussion.

In Figure 7:

The overall analysis of this part is rather poor and may not qualify the level of requirement of our journal. I feel that the rest is already sufficient to confirm that dimerization deficiency of mammalian NotchICD can cause pathology in the gut, heart, and B cells. Therefore, I suggest to remove this part from the manuscript unless some of the issues below are addressed:

1. The Fig S6A doesn’t look consistent among different replicates. S6B also show high level of variation, so it is extremely difficult to make any conclusion.

2. The RNAseq analysis with N=3 has failed in showing consistency within each group by having an outlier. Unless it has clear correlation to the severity of disease phenotype of the sacrificed mouse, it seems very difficult to make any conclusion here. Fig7D could be used to show the penetrance of increased expression of the selected genes – number of samples for each genotype is not shown and here many mice would need to be tested to make a firm conclusion.

3. Given the lack of solid evidence that Myb and FoxM1 are involved here, further chromosomal analysis with ATAC seq and SplitDAM (although I greatly appreciate the methods) doesn’t seem yield any solid conclusion. This all might need further in-depth study to reveal the true molecular mechanism of the observed phenotypes (for all three main phenotypes).

In conclusion, the suggestion would be to finish the manuscript at Fig 6 and Fig S5. Fig 8 might be included but after revising the contents. Perhaps a different schematic drawing to show dimerization deficiency mechanism and the affected pathological aspects in mouse would work better.

---

## [Editor Report · Decision Letter 3]

15 Jul 2020

Dear Dr Kopan,

Thank you for submitting your revised Research Article entitled "Exposure to Mites Sensitizes Intestinal Stem Cell Maintenance, Splenic Marginal Zone B Cell Homeostasis, And Heart Development to Notch Dosage and Cooperativity." for publication in PLOS Biology. I have now discussed the revision with the other editors and obtained advice from the Academic Editor. 

We're delighted to let you know that we're now editorially satisfied with your manuscript. The only change we would like you to do in the manuscript is to revise the title to make it more accessible and we have come up with two alternatives. Please choose one of them and change the title whenever you are ready to submit the final version of the manuscript:

1) Notch dimerization is important for normal heart development, intestinal stem cell maintenance and splenic marginal zone B cell homeostasis.

2) Notch dimerization and gene dosage are important for normal heart development, intestinal stem cell maintenance and splenic marginal zone B cell homeostasis during mite infestation 

Before we can formally accept your paper and consider it "in press", we also need to ensure that your article conforms to our guidelines. A member of our team will be in touch shortly with a set of requests. As we can't proceed until these requirements are met, your swift response will help prevent delays to publication. Please also make sure to address the data and other policy-related requests noted at the end of this email.

*Copyediting*

*Published Peer Review History*

*Early Version*

*Submitting Your Revision*

Best wishes,

Ines

--

Ines Alvarez-Garcia, PhD

Senior Editor

PLOS Biology

Carlyle House, Carlyle Road

Cambridge, CB4 3DN

+44 1223–442810

DATA POLICY:

Many thanks for adding the data file containing the raw data underlying all the graphs shown in the main and supplementary figures. Please amend the following:

- Rename the file 'S1_Data'

- In the data from Fig. 2, relabel data shown in Fig. 2J for 2H (there is no J in the figure).

- As you have to relabel the file, please amend in each of the corresponding figure legends the sentence indicating where the data can be found and add this also to each of the corresponding legends of the supplementary files.

- You have indicated that you will submit to GEO all the molecular data. Please do so before acceptance to production and indicate the GEO number in the Data Availability section.

- There is an instance of 'data not shown' in page 10. Please either provide the data in the supplementary figures or remove the statement. All data must be shown.

While you have added some of them to the data file, others seem to be missing. Please provide them all in the same file.

---

## [Editor Report · Decision Letter 4]

2 Sep 2020

Dear Dr Kopan,

On behalf of my colleagues and the Academic Editor, Bon-Kyoung Koo, I am pleased to inform you that we will be delighted to publish your Research Article in PLOS Biology. 

Early Version

PRESS 

Kind regards,

Alice Musson

Publishing Editor, 

PLOS Biology

on behalf of

Ines Alvarez-Garcia,

Senior Editor

PLOS Biology